# LEARNING VORTEX DYNAMICS FOR FLUID INFERENCE AND PREDICTION

**Yitong Deng**
Dartmouth College
Stanford University

**Hong-Xing Yu**
Stanford University

**Jiajun Wu**
Stanford University

**Bo Zhu**
Dartmouth College

## ABSTRACT

We propose a novel *differentiable vortex particle* (DVP) method to infer and predict fluid dynamics from a single video. Lying at its core is a particle-based latent space to encapsulate the *hidden*, Lagrangian vortical evolution underpinning the *observable*, Eulerian flow phenomena. Our *differentiable vortex particles* are coupled with a learnable, vortex-to-velocity dynamics mapping to effectively capture the complex flow features in a physically-constrained, low-dimensional space. This representation facilitates the learning of a fluid simulator tailored to the input video that can deliver robust, long-term future predictions. The value of our method is twofold: first, our learned simulator enables the inference of hidden physics quantities (*e.g.*, velocity field) purely from visual observation; secondly, it also supports future prediction, constructing the input video's *sequel* along with its future dynamics evolution. We compare our method with a range of existing methods on both synthetic and real-world videos, demonstrating improved reconstruction quality, visual plausibility, and physical integrity.[1]

## 1   INTRODUCTION

As small as thin soap films, and as large as atmospheric eddies observable from outer space, fluid systems can exhibit intricate dynamic features on different mediums and scales. However, despite recent progress, it remains an open problem for scientific machine learning to effectively represent these flow features, identify the underlying dynamics system, and predict the future evolution, due to the noisy data, imperfect modeling, and unavailable, *hidden* physics quantities.

Here, we identify three fundamental challenges that currently hinder the success of such endeavors. First, *flow features are difficult to represent*. Traditional methods learn the fluid dynamics by storing velocity fields either using regularly-spaced grids or smooth neural networks. These approaches have demonstrated promising results for fluid phenomena that are relatively damped and laminar (*e.g.*, Chu et al., 2022), but for fluid systems that can exhibit turbulent features on varying scales, these methods fall short due to the problem's curse of dimensionality (high-resolution space and time), local non-smoothness, and hidden constraints. As a result, more compact and structured representation spaces and data structures are called for.

Secondly, *hidden flow dynamics is hard to learn*. Fluid systems as prescribed by the Navier-Stokes equations tightly couple multiple physical quantities (*i.e.*, velocity, pressure, and density), and yet, only the density information can be accessibly measured. Due to the system's complexity, ambiguity, and non-linearity, directly learning the underlying dynamics from the observable density space is infeasible; and successful learning usually relies on velocity or pressure supervision, a requirement that distances these methods from deployment in real-world scenarios.

Exciting recent progress has been made in hidden dynamics inference by PDE-based frameworks such as Raissi et al. (2020), which uncover the underlying physics variables solely from density observations. However, this type of methods encounter the third fundamental challenge, which is that *performing future prediction is difficult*. As we will demonstrate, although strong results

---

[1]Our video results, code, and data can be found at our project website: `https://yitongdeng.github.io/vortex_learning_webpage`.

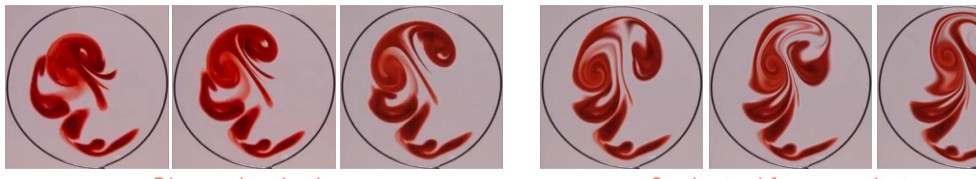

Figure 1: Our goal is to learn vortex dynamics for fluid inference and prediction. The 3 frames on the left are observed from a real video of a soap film on a circular metal rim, where the red ink is spreading. The 3 frames on the right are future prediction results produced by our method.

are obtained for interpolating inside the observation window provided by the training data, these methods cannot extrapolate into the future, profoundly limiting their usage.

In this paper, we propose the *differentiable vortex particle* (DVP) method, a novel fluid learning model to tackle the three aforementioned challenges in a unified framework. In particular, harnessing the physical insights developed for the *vortex methods* in the computational fluid dynamics (CFD) literature, we design a novel, data-driven Lagrangian vortex system to serve as a compact and structured *latent representation* of the flow dynamics. We learn the complex dynamical system underneath the high-dimensional image space by learning a surrogate, low-dimensional model on the latent vortex space, and use a physics-based, learnable mechanism: the vortex-to-velocity module, to *decode* the latent-space dynamics back to the image space. Leveraging this physics-based representation, we design an end-to-end training pipeline that learns from a single video containing only density information. Our DVP method enables accurate inference of hidden physics quantities and robust long-term future predictions, as shown in Figure 1.

To examine the efficacy of our method, we compare our method's performance on both *motion inference* and *future prediction* against various state-of-the-art methods along with their extensions. We conduct benchmark testing on synthetic videos generated using high-order numeric simulation schemes as well as real-world videos in the wild. Evaluation is carried out both quantitatively through exhaustive numerical analysis, and qualitatively by generating a range of realistic visual effects. We compare the uncovered velocities in terms of both reconstruction quality and physical integrity, and the predicted visual results in terms of both pixel-level and perceptual proximity. Results indicate that our proposed method provides enhanced abilities on both fronts, inferring hidden quantities at higher accuracy, and predicting future evolution with higher plausibility.

In summary, the main technical contributions of our framework align with the three challenges regarding flow representation, dynamics learning, and simulator synthesis. (1) We devise a novel representation for fluid learning, the *differentiable vortex particles* (DVP), to drastically reduce the learning problem's dimensionality on complex flow fields. Motivated by the vortex methods in CFD, we establish the vorticity-carrying fluid particles as a new type of learning primitive to transform the existing PDE-constrained optimization problem to a particle trajectory (ODE) learning problem. (2) We design a novel particle-to-field paradigm for learning the Lagrangian vortex dynamics. Instead of learning the interaction among particles (*e.g.*, Sanchez-Gonzalez et al., 2020), our model learns the continuous vortex-to-velocity induction mapping to naturally connect the vortex particle dynamics in the latent space with the fluid phenomena captured in the image space. (3) We develop an *end-to-end differentiable pipeline* composed of two network models to synthesize data-driven simulators based on single, short RGB videos.

## 2 RELATED WORK

*Hidden Dynamics Inference.* The problem of inferring dynamical systems based on noisy or incomplete observations has been addressed using a variety of techniques, including symbolic regression (Bongard & Lipson, 2007; Schmidt & Lipson, 2009), dynamic mode decomposition (Schmid, 2010; Kutz et al., 2016), sparse regression (Brunton et al., 2016; Rudy et al., 2017), Gaussian process regression (Raissi et al., 2017; Raissi & Karniadakis, 2018), and neural networks (Raissi et al., 2019; Yang et al., 2020; Jin et al., 2021; Chu et al., 2022). Among these inspiring advancements, the "hidden fluid mechanics" (HFM) method proposed in Raissi et al. (2020) is particularly noteworthy, as it uncovers the continuous solutions of fluid flow using only images (the transport of smoke or ink).

*Data-driven Simulation.* Recently, growing interests are cast on learning numerical simulators according to data supervision, which has shown promise to reduce computation time (Ladický et al.,

2015; Guo et al., 2016; Wiewel et al., 2019; Pfaff et al., 2020; Sanchez-Gonzalez et al., 2020; Tompson et al., 2017), increase simulation realism (Chu & Thuerey, 2017; Xie et al., 2018), enable stylized control (Kim et al., 2020), estimate dynamic quantities such as viscosity and energy (Chang et al., 2016; Battaglia et al., 2016; Ummenhofer et al., 2019), and facilitate the training of control policies (Sanchez-Gonzalez et al., 2018; Li et al., 2018). Akin to Watters et al. (2017), our system takes images as inputs and performs dynamics simulation on a low-dimensional latent space; but our method learns purely from the input video and performs future rollout in the image space. Our method is also related to Guan et al. (2022), which infers Lagrangian fluid simulation from observed images. We propose sparse neural vortices as our representation while they use dense material points.

*Vortex Methods.* The underlying physical prior incorporated in our machine learning system is rooted in the family of vortex methods that are rigorously derived, analyzed, and tested in the computational fluid dynamics (CFD) (Leonard, 1980; Perlman, 1985; Beale & Majda, 1985; Winckelmans & Leonard, 1993; Mimeau & Mortazavi, 2021) and computer graphics (CG) (Selle et al., 2005; Park & Kim, 2005; Weißmann & Pinkall, 2010; Brochu et al., 2012) communities. Xiong et al. (2020) is pioneering for combining the Discrete Vortex Method with neural networks, but its proposed method relies on a large set of ground truth velocity sequences, whereas our method learns from single videos without needing the ground truth velocity.

## 3 PHYSICAL MODEL

We consider the velocity-vorticity form of the Navier–Stokes equations (obtained by taking the curl operator of both sides of the momentum equation, see Cottet et al. (2000) for details):

$$\frac{D\boldsymbol{\omega}}{Dt} = \frac{\partial \boldsymbol{\omega}}{\partial t} + \boldsymbol{u} \cdot \nabla \boldsymbol{\omega} = \boldsymbol{\omega} \cdot \nabla \boldsymbol{u} + \nu \nabla^2 \boldsymbol{\omega} + \nabla \times \boldsymbol{b}, \tag{1}$$

$$\boldsymbol{u} = \nabla \times \boldsymbol{\phi}, \quad \nabla^2 \boldsymbol{\phi} = -\boldsymbol{\omega}, \tag{2}$$

where $\boldsymbol{\omega}$ denotes the vorticity, $\boldsymbol{u}$ the velocity, $\boldsymbol{b}$ the conservative body force, $\nu$ the kinematic viscosity, and $\phi$ the streamfunction. If we ignore the viscosity and stretching terms (inviscid 2D flow), we obtain $D\boldsymbol{\omega}/Dt = \boldsymbol{0}$, which directly conveys the Lagrangian conservative nature of vorticity (*i.e.*, a particle's vorticity will not change during its advection).

If we assume the fluid domain has an open boundary, we can further obtain the vortex-to-velocity induction formula, which is derived by solving Poisson's equation on $\phi$ using Green's method (also known as the Biot-Savart Law in fluid mechanics):

$$\boldsymbol{u}(\boldsymbol{x}) = \int K(\boldsymbol{x} - \boldsymbol{x}')\boldsymbol{\omega}(\boldsymbol{x}')d\boldsymbol{x}', \tag{3}$$

The kernel $K$ exhibits a type-II singularity at $\boldsymbol{0}$ and causes numerical instabilities, therefore in CFD practices, $K$ is replaced by various mollified versions $K_\delta$ to improve the simulation accuracy (Beale & Majda, 1985). We note that the mollified version $K_\delta$ is not unique, and can be customized and tuned in different numerical schemes per human heuristics. Different types and parameters for the mollification bring about significantly different simulation results.

*Takeaways.* The mathematical models above provide two central physical insights guiding the design of our vortex-based learning framework: (1) The Lagrangian conservation of vorticity $\boldsymbol{\omega}$ suggests the suitability of adopting Lagrangian data structures (*e.g.*, particles as opposed to grids) to capture the dynamics. Since the tracked variable $\boldsymbol{\omega}$ remains temporally invariant for each Lagrangian vortex, the evolution of the continuous flow field is embodied fully by the movement of these vortices, which significantly alleviates the difficulty in learning. (2) Equation 3 presents an induction mapping from the vorticity $\boldsymbol{\omega}$, a Lagrangian quantity carried by particles, to the velocity $\boldsymbol{u}$, an Eulerian variable that can be queried continuously at an arbitrary location $\boldsymbol{x}$. This lends the possibility for the Lagrangian method to be used in conjunction with Eulerian data structures (*e.g.*, a grid) for learning from the widely available video data. Furthermore, such a mapping can benefit from data-driven learning, as we can replace the human heuristics by learning a mollified kernel $K_\delta$ to minimize the discrepancy between the simulated and observed flow phenomena.

## 4 METHOD

*System Overview.* Following the physics insight conveyed in Section 3, we design a learning system whose workflow is illustrated in Figure 2. As shown on the top row, our system takes as input a

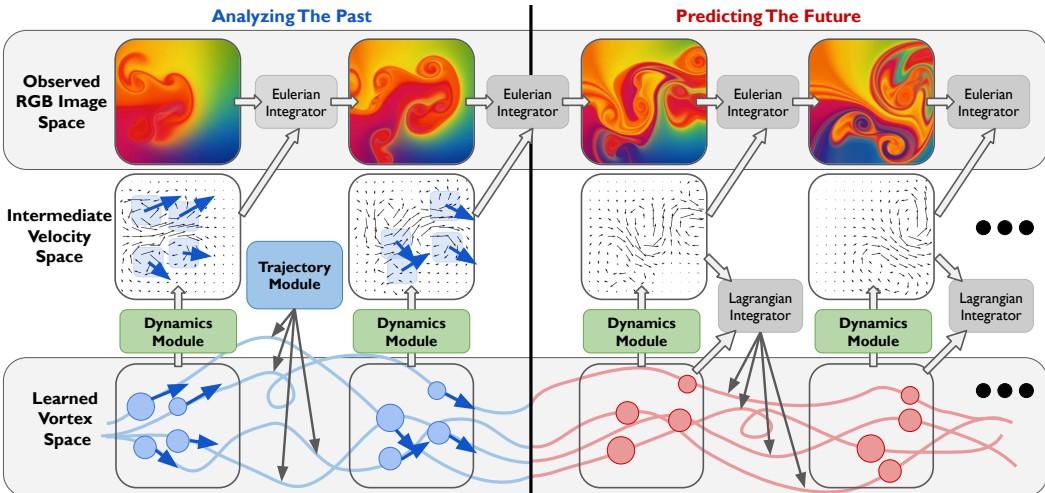

Figure 2: Illustration of our differentiable vortex particle (DVP) method. Given an input RGB image sequence (top row), we learn a dynamical system on a low-dimensional vortex space (bottom row), whose motion is decoded into the motion of the high-dimensional image space to explain the observed fluid phenomena.

single RGB video that captures the vortical flow phenomena. As shown on the bottom row, our method learns and outputs a dynamical simulator — not on the image space itself, but on a latent space consisting of discrete vortices. Learning the latent dynamics in the vortex space would only be useful and feasible if we can tie it back to the image space, because it is the image space that we want to perform future prediction on, and we have no ground truth values for the vortex particles to begin with. The bridge to tie the vortex space with the image space derives from Equation 3, which supplies the core insight that there exists a learnable mapping from vortex particles to the continuous velocity field at arbitrary positions. This mapping is modeled by our learned dynamics module $\mathcal{D}$, which gives rise to the intermediate velocity space, as shown in the middle row of Figure 2.

## 4.1 DIFFERENTIABLE VORTEX PARTICLES

We track a collection $\mathcal{V}$ of $n$ vortex particles, *i.e.*, $\mathcal{V} \coloneqq [V_1, \ldots, V_n]$. We define each vortex $V_i$ as the 3-tuple $(\boldsymbol{x}_i, \omega_i, \delta_i)$, where $\boldsymbol{x}$ represents the position, $\omega$ the vortex strength, and $\delta$ the size. The number of particles $n$ is a hyperparameter which we set to 16 for all our results. Further discussions and experiments regarding the choice of $n$ can be found in Appendix D. We also note that, since we are concerned with 2D inviscid incompressible flow, the size $\delta$ of a vortex does not change in time due to incompressibility, and the vortex strength $\omega$ does not change in time due to Kelvin's circulation theorem (see Hald (1979) for a thorough discussion).

*Learning Particle Trajectory.* As shown in Figure 3, we learn a particle trajectory module: a query function $\mathcal{T}$ such that $\mathcal{V}_t = \mathcal{T}(t)$, which predicts the configuration of all the vortices at any time $t \in [0, t_E]$ where $t_E$ represents the end time of the input video. As described above, predicting $\mathcal{V}_t$ boils down to determining two time-invariant components: (1) $[\omega_1, \ldots, \omega_n]$, (2) $[\delta_1, \ldots, \delta_n]$, and one time-varying component: $[(\boldsymbol{x}_1)_t, \ldots, (\boldsymbol{x}_n)_t]$. For the two time-invariant components, we introduce two trainable $n \times 1$ vectors $\Delta$ and $\Omega$ to represent $\delta$ and $\omega$ respectively, such that $[\omega_1, \ldots, \omega_n] = \sin(\Omega)$ and $[\delta_1, \ldots, \delta_n] = \text{sigmoid}(\Delta) + \epsilon$ ($\epsilon$ is a hyperparameter we set to 0.03). The vortex size $\Delta$ and strength $\Omega$ are optimized to fit the motion depicted by the input RGB video. For the time-varying component, we use a neural network $N_1(t)$ to encode $N_1(t) = [(\boldsymbol{x}_1)_t, \ldots, (\boldsymbol{x}_n)_t]$, and the particle velocities $\frac{dN_1}{dt}$ can be extracted using automatic differentiation. We note that learning the full particle trajectory, rather than the initial particle configuration, allows the aggregation of dynamics information throughout the input video for better inference and prediction. We provide further discussion on this design in Appendix F.

*Trajectory Initialization.* As discussed above, the trajectory $\mathcal{T}$ has three learnable components: $\Delta$, $\Omega$ and $N_1$. We initialize $\Delta$ and $\Omega$ as zero vectors, which gives $\delta_i = 0.5 + \epsilon$ and $\omega_i = 0$ for all $i$. Conceptually, these vortices are initialized as large blobs with no vortex strength, which learn to alter their sizes and grow their strengths to better recreate the eddies seen in the video. The initial positions $[(\boldsymbol{x}_1)_0, \ldots, (\boldsymbol{x}_n)_0]$ are regularly spaced points to populate the entire domain. We initialize

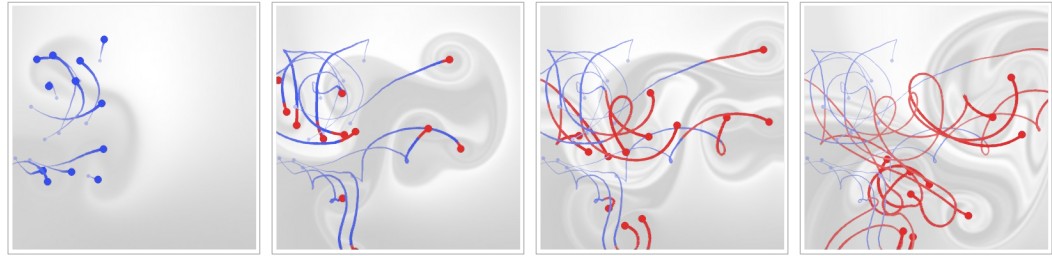

Figure 3: We encapsulate the motion of a continuous field by the motion of discrete particles. The blue trajectory is encoded by a neural network $N_1$, corresponding to the input video; while the red trajectory is unrolled using our learned dynamics module and a numeric integrator, corresponding to the future prediction.

the 16 particles to lie at the grid centers of a $4 \times 4$ grid. To do so, we simply pretrain $N_1$ so that $N_1(0)$ evaluates to the grid centers. The details regarding pretraining are given in Appendix A.

*Learning the Vortex-to-Velocity Mapping.* The vortex-to-velocity mapping is performed by our dynamics module $\mathcal{D}$, which predicts the velocity $\boldsymbol{u}$ at an arbitrary query point $\boldsymbol{x}$ given the collection of vortices $\mathcal{V} = [(\boldsymbol{x}_1, \omega_1, \delta_1), \ldots, (\boldsymbol{x}_n, \omega_n, \delta_n)]$. Following the physical insight conveyed in Section 3, $\mathcal{D}$ should evaluate the integration:

$$\boldsymbol{u}(\boldsymbol{x}) = \int K_\delta(\boldsymbol{x} - \boldsymbol{x}')\boldsymbol{\omega}(\boldsymbol{x}')d\boldsymbol{x}', \tag{4}$$

which replaces the kernel $K$ in Equation 3 by a learnable mapping $K_\delta : \mathbb{R}^d \to \mathbb{R}^d$, with $d$ representing the spatial dimension. Rather than directly using a neural network to model this $\mathbb{R}^d \to \mathbb{R}^d$ mapping, we incorporate further physical insights by analyzing the structure of $K_\delta$. As derived in Beale & Majda (1985), the kernel $K_\delta$ for 2-dimensional flow exhibits the following form:

$$K_\delta(\boldsymbol{z}) = \frac{1}{2\pi r}M(r, \delta)R_{\frac{2}{\pi}}(\boldsymbol{z}), \; r = |\boldsymbol{z}| \tag{5}$$

where $R_{\frac{2}{\pi}}(\boldsymbol{z})$ computes the unit direction of the cross product of $\boldsymbol{z}$ and the out-of-plane unit vector; and $M(r, \delta)$ is the human heuristic term that varies by choice. Hence, we opt to replace $\frac{1}{2\pi r}M(r, \delta)$ by a $\mathbb{R}^2 \to \mathbb{R}$ neural network function $N_2(r, \delta)$ so that:

$$\boldsymbol{u}(\boldsymbol{x}) = \int N_2(|\boldsymbol{x} - \boldsymbol{x}'|, \delta_i)R_{\frac{2}{\pi}}(\boldsymbol{x} - \boldsymbol{x}')\boldsymbol{\omega}(\boldsymbol{x}')d\boldsymbol{x}' \tag{6}$$

$$\approx \sum_{i=1}^{n} N_2(|\boldsymbol{x} - \boldsymbol{x}_i|, \delta_i)R_{\frac{2}{\pi}}(\boldsymbol{x} - \boldsymbol{x}_i)\omega_i = \mathcal{D}(\mathcal{V})(\boldsymbol{x}). \tag{7}$$

Learning this induction kernel $N_2(r, \delta)$ instead of using heuristics-based kernels allows for more accurate fluid learning and prediction from input videos. We discuss more on this in Appendix E.

## 4.2 END-TO-END TRAINING

As previously mentioned, the dynamics on the latent vortex space is bridged to the evolution of the image space through the differentiable, dynamics module $\mathcal{D}$. Hence, we can optimize the vortex representation $\mathcal{V}_t = \mathcal{T}(t)$ at time $t$ using images as supervision. First, we select $m + 1$ frames: $[I_t, \ldots, I_{t+m}]$ from the video. Then, we compute $\boldsymbol{u}_t = \mathcal{D}(\mathcal{V}_t)$. After that, $(\boldsymbol{u}_t, I_t)$ is fed into an integrator on the Eulerian grid to predict $\tilde{I}_{t+1}$. Simultaneously, $(\boldsymbol{u}_t, \mathcal{V}_t)$ is fed into an integrator on the Lagrangian particles to predict $\tilde{\mathcal{V}}_{t+1}$. The process is then repeated, using $\tilde{I}_{t+1}$ in place of $I_t$ and $\tilde{\mathcal{V}}_{t+1}$ in place of $\mathcal{V}_t$, to generate $\tilde{I}_{t+2}$ and $\tilde{\mathcal{V}}_{t+2}$, and so on. Eventually, we would obtain $[\tilde{I}_{t+1}, \ldots, \tilde{I}_{t+m}]$, which are the predicted outcome starting at time $t$. We optimize $\mathcal{T}$ and $\mathcal{D}$ jointly by minimizing the difference between $[\tilde{I}_{t+1}, \ldots, \tilde{I}_{t+m}]$ and $[I_{t+1}, \ldots, I_{t+m}]$ in an end-to-end fashion.

By picking different values of $t$ in each training iteration to cover $[0, t_E]$, we optimize $\mathcal{T}$ and $\mathcal{D}$ to fit the input video. There remains one more caveat — which is that the trajectories encoded with $\mathcal{T}$ are not enforced to be consistent with $\mathcal{D}$, because each frame of $\mathcal{V}_t$ is optimized individually. In other words, if we evaluate the particle velocities $[\dot{\boldsymbol{x}}_1, \ldots, \dot{\boldsymbol{x}}_n] = \frac{dN_1}{dt}$ as prescribed by $\mathcal{T}$, it

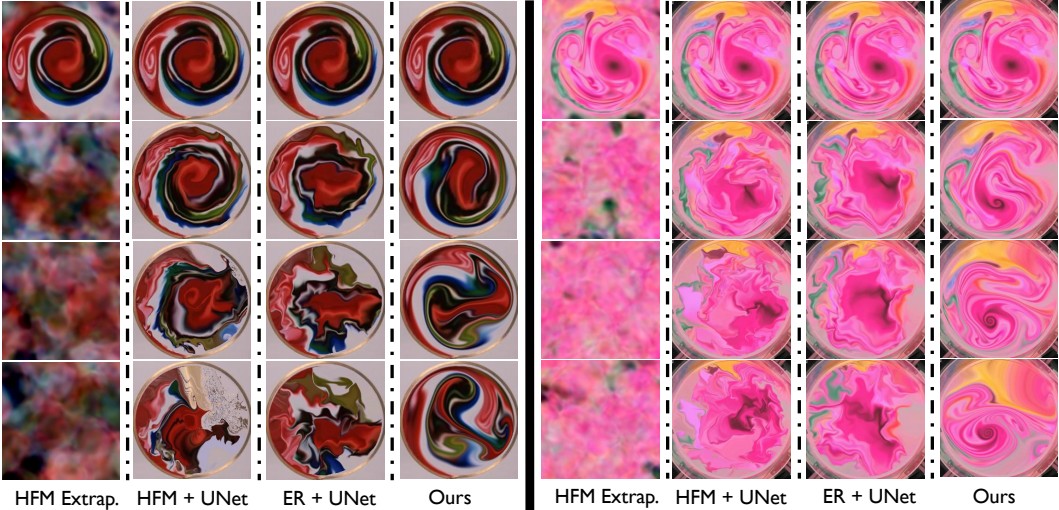

Figure 4: Applied to real-world videos, our DVP method can create more realistic future predictions over long periods of time compared to existing methods (and their extensions).

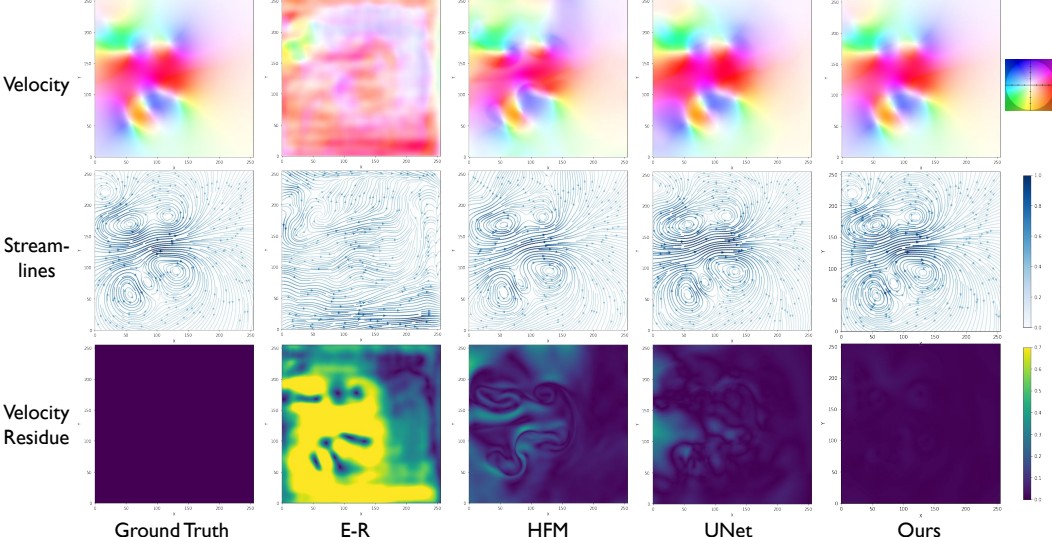

Figure 5: Hidden motion inference compared with existing methods on a synthetic video. Our DVP method uncovers the underlying velocity field at higher accuracy.

should coincide with $[\mathcal{D}(\mathcal{V})(\boldsymbol{x_1}), \ldots, \mathcal{D}(\mathcal{V})(\boldsymbol{x_n})]$, as prescribed by $\mathcal{D}$. Hence, in training, another loss is computed between $\frac{dN_1}{dt}$ and $[\mathcal{D}(\mathcal{V})(\boldsymbol{x_1}), \ldots, \mathcal{D}(\mathcal{V})(\boldsymbol{x_n})]$ to align the vortex trajectory and the predicted velocity.

*Deployment.* After successful training, our learned system performs two important tasks. First, using our query function $\mathcal{T}(t)$, we are able to temporally interpolate for $\mathcal{V}_t$, which then uncovers the hidden velocity field $\boldsymbol{u} = \mathcal{D}(\mathcal{V}_t)$ at arbitrary resolutions, providing the same functionality as Raissi & Karniadakis (2018), but using vorticity instead of pressure as the secondary variable. Moreover, with the dynamics module $\mathcal{D}$, we can perform future prediction to unroll the input video, a feature unsupported by previous methods. As shown in Figure 4, since our method is forward-simulating by nature, it can provide more realistic and robust future predictions than existing methods or their extensions. Further implementation details of our method, including hyperparameters, network architectures, training schemes, and computational costs can be found in Appendix A.

## 5 EXPERIMENTS

We evaluate our method's ability to perform motion inference and future prediction on both synthetic and real videos, comparing against existing methods.

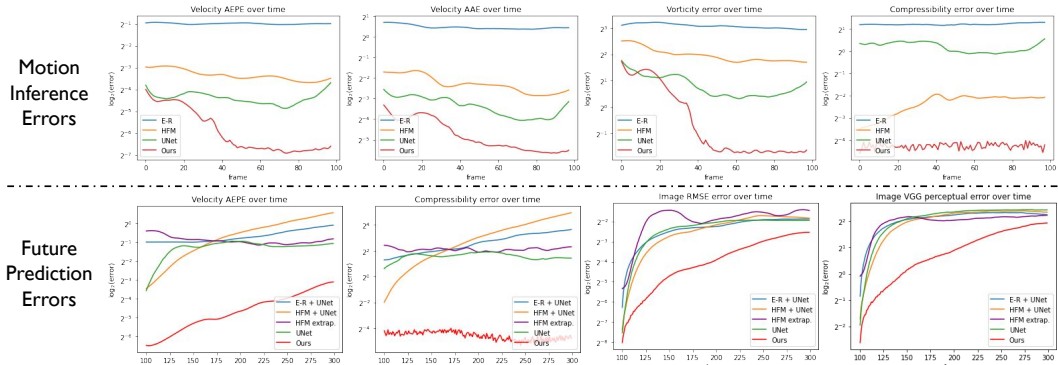

Figure 6: Error analysis on a synthetic video. The top row plots the inference errors of velocity, vorticity, and compressibility. The bottom row plots the future prediction errors, which consider both the dynamics error of the velocity and the perceptual error of the generated image sequence.

*Baselines.* For motion inference, we compare our method against Raissi & Karniadakis (2018) (HFM) and Zhang et al. (2022) (E-R). We reimplement the HFM method as prescribed in the paper, making only the modification that instead of using only a single concentration variable $c$ and its corresponding variable $d := 1 - c$, we create three $(c, d)$ pairs for each of the RGB channel for the support of colored videos. The E-R method is evaluated using the published pretrained models. We further compare against an ablated version of our proposed method, termed "UNet", which essentially replaces the Lagrangian components of the system with a UNet architecture (Ronneberger et al., 2015), a classic method for learning field-to-field mappings. The UNet baseline takes two images $I_t$ and $I_{t+1}$ and predicts a velocity field $\boldsymbol{u}_{t+1}$ to predict $I_{t+2}$ using the same Eulerian integrator as our method. For future prediction, there do not exist previous methods that operate in comparable settings, so we extend the inference methods in a few ways to support future prediction in a logical and straightforward manner. First, since HFM offers a query function parameterized by $t$, we test its future prediction behavior by simply extrapolating with $t > t_E$; this is referred to as "HFM extp.". Since both Raissi & Karniadakis (2018) and Zhang et al. (2022) uncover the time-varying velocity field, we use a UNet to learn the evolution from $\boldsymbol{u}_t$ to $\boldsymbol{u}_{t+1}$, and use this velocity update mechanism to perform future prediction. The two baselines thus obtained are referred to as "HFM+UNet" and "E-R+UNet" respectively. Our method's ablation "UNet" supports future prediction intrinsically.

## 5.1 SYNTHETIC VIDEO

The synthetic video for vortical flow is generated using the Discrete Vortex Method with a first-order Gaussian mollifying kernel $M$ (Beale & Majda, 1985). The high-fidelity BFECC advection scheme (Kim et al., 2005) with Runge-Kutta-3 time integration is deployed. The simulation advects a background grid of size $256 \times 256$, with a time step $dt = 0.01$ to create 300 simulated frames. Only the first 100 frames will be disclosed to train all methods, and future predictions are tested and examined on the following 200 frames.

*Motion Inference.* The results for the uncovering of hidden dynamic variables are illustrated in Figure 5 and Figure 6. Shown in Figure 5 are the velocities uncovered by all 4 methods against the ground truth, at frame 55 of the synthetic video with 100 observed frames. The velocity is visualized in the forms of colors (top row) and streamlines (middle row), while the velocity residue, measured in end-point error (EPE), is depicted on the bottom row. It can be seen that HFM, UNet, and our method achieve agreeing results, all matching the ground truth values at high accuracy. On the bottom row, it can be seen that as compared to HFM and UNet, our method generates the inference velocity that best matches the unseen ground truth.

The inference results over the full 100 frames are depicted at the top of Figure 6. We evaluate the velocity with four metrics: the average end-point error (AEPE), average angular error (AAE), vorticity RMSE and compressibility RMSE. From all 4 metrics, it can be seen that our method outperforms the baselines consistently. The time-averaged data for all four metrics are shown on the left of Table 1, which deems our method favorable for all metrics used.

*Future Prediction.* In Figure 7, we visually compare the future prediction results (from frame 100 to frame 299) using our method and the 4 benchmarks against the ground truth. It can be seen that the

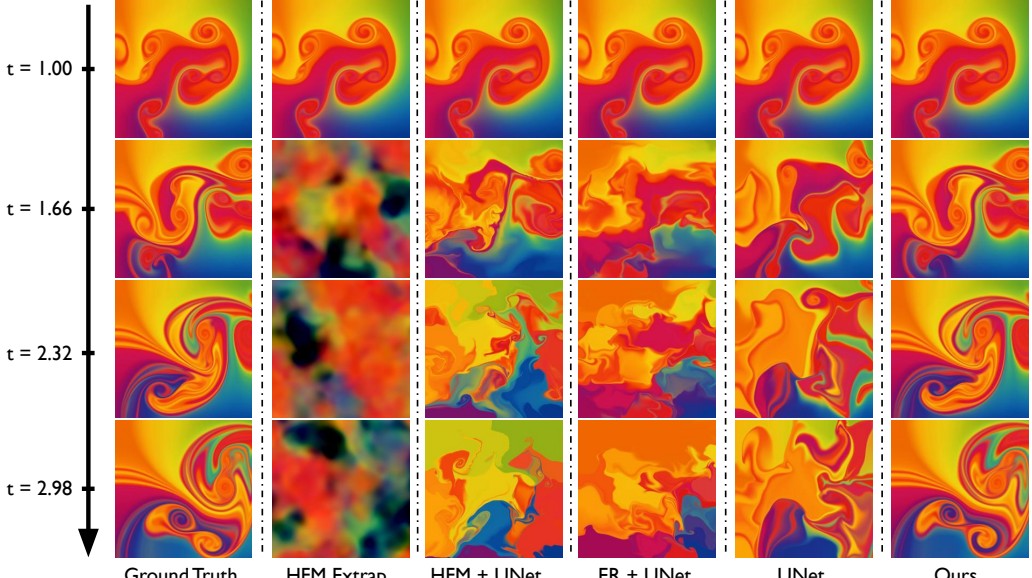

Figure 7: Future prediction of our DVP method compared to baselines methods. Our method accurately predicts the unseen, future sequence that is twice as long as the seen sequence.

| | Time-averaged Inference Errors | | | | | Time-averaged Prediction Errors | | | | | |
|---|---|---|---|---|---|---|---|---|---|---|---|
| | AEPE | AAE | Vort. | Div. | | VGG | RMSE | AEPE | AAE | Vort. | Div. |
| E-R | 0.505 | 1.393 | 8.470 | 2.319 | +UNet | 4.346 | 0.205 | 0.631 | 1.424 | 12.84 | 6.580 |
| HFM | 0.100 | 0.212 | 3.949 | 0.202 | +UNet | 4.258 | 0.205 | 0.720 | 1.062 | 36.73 | 10.41 |
| | | | | | Extp. | 4.080 | 0.285 | 0.541 | 1.464 | 7.761 | 4.315 |
| UNet | 0.048 | 0.100 | 1.799 | 1.145 | | 4.530 | 0.211 | 0.424 | 1.159 | 7.334 | 3.017 |
| Ours | **0.020** | **0.041** | **0.976** | **0.053** | | **2.010** | **0.080** | **0.048** | **0.096** | **1.621** | **0.043** |

Table 1: Error analysis of benchmark testing on a synthetic dataset.

sequence generated by our method best matches the ground truth video, capturing the vortical flow structures, while the other baselines either quickly diffuse or generate unnatural, hard-edged patterns. Numerical analysis confirms these visual observations, as we compare the 200 future frames in terms of both velocity and visual similarity. The velocity analysis inherits the same 4 metrics, and the visual similarity is gauged using the pixel-level RMSE and the VGG feature reconstruction loss (Johnson et al., 2016). The time-averaged results of all 6 metrics are documented on the right of Table 1, and the time-varying results are plotted on the bottom of Figure 6. It can be concluded from the visual and numerical evidence that our method outperforms the baselines in this case.

## 5.2 REAL VIDEO

A similar numerical analysis is carried out on a real video published on YouTube, as shown in Figure 8. The video has 150 frames: the first 100 frames will be used for training, while the remaining 50 frames will be reserved for testing. Since the ground truth velocities for the real video are intractable, we will only analyze the future-predicting performance. For all methods, we perform future prediction for 150 frames; among these, the first 50 frames will be compared with the original video, and the rest (100 frames) will be evaluated visually and qualitatively. Since only part of the video is fluid (within the circular rim), we pre-generate a signed distance field for all methods, so that only the fluid regions are considered in learning and simulation. The same boundary condition is employed for all methods (except for "HFM exp." which requires no advection).

The numerical analysis for the first 50 predicted frames is documented and plotted in Table 2 and Figure 9. We compare our method against the baselines based on the VGG perceptual loss for visual plausibility, and the velocity divergence (which should in theory be 0 for incompressible fluid) for physical integrity. It can be seen that our method prevails on all metrics used. For prediction

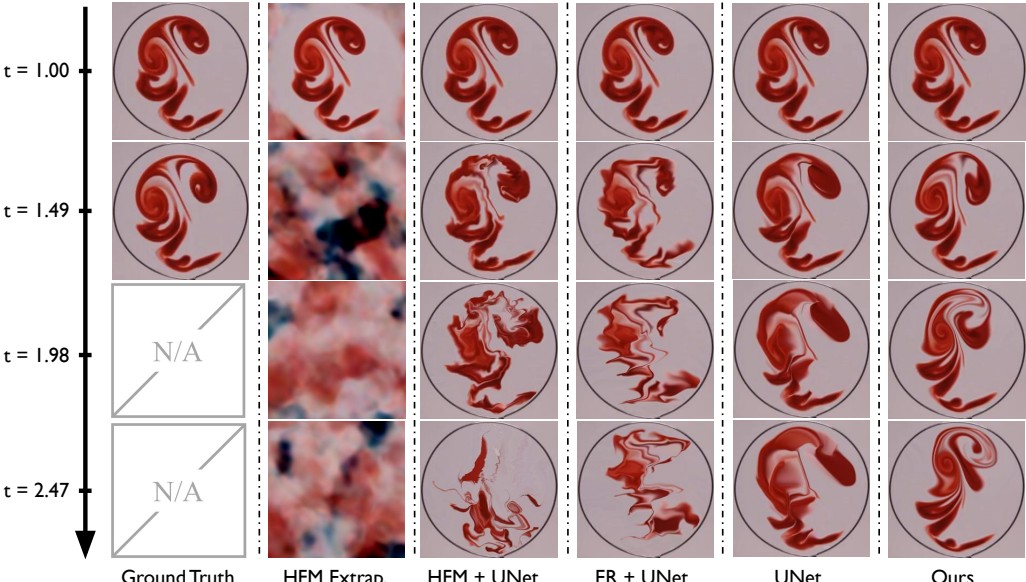

Figure 8: Future prediction of our DVP method compared to baseline methods on a real video sequence. Our method generates a predicted sequence that best matches the input video within its duration, and remains visually plausible way beyond its duration.

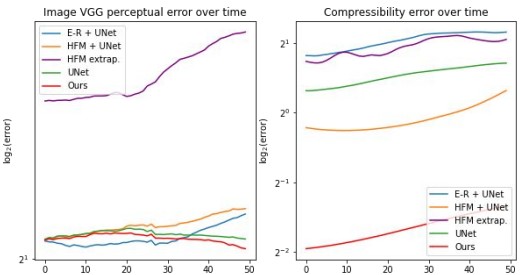

Figure 9: Error plots corresponding to Figure 8.

|  | VGG (avg.) | VGG (final) | Div. (avg) |
|---|---|---|---|
| E-R | 2.095 | 2.205 | 2.046 |
| HFM+UNet | 2.151 | 2.231 | 0.940 |
| HFM Extp. | 2.980 | 3.271 | 1.922 |
| UNet | 2.111 | 2.088 | 1.447 |
| Ours | **2.093** | **2.045** | **0.318** |

Table 2: Time-averaged errors for Figure 8.

results that exceed the duration of the real video, qualitative observations can be made: our method preserves the vortical structures and generates smooth visualizations over the entire time horizon, while other methods end up yielding glitchy patterns.

We perform additional quantitative benchmark testings in Appendix B against a differentiable grid-based simulator on real and synthetic videos; and in Appendix C against 4 baselines on another synthetic video featuring different visual and dynamical distributions.

## 6  CONCLUSION & LIMITATIONS

In this work, we propose a novel data-driven system to perform fluid hidden dynamics inference and future prediction from single RGB videos, leveraging a novel, vortex latent space. The success of our method in synthetic and real data, both qualitatively and quantitatively, suggests the potential for embedding Lagrangian structures for fluid learning. Our method has several limitations. First, our vortex model is currently limited to 2D inviscid flow. Extending to 3D, viscous flow is an exciting direction, which can be enabled by allowing vortex strengths and sizes to evolve in time (Mimeau & Mortazavi, 2021). Secondly, our vortex evolution did not take into account the boundary conditions in a physically-based manner, hence it cannot accurately predict flow details around a solid boundary. Incorporating learning-based boundary modeling may be an interesting exploration. Thirdly, scaling our method to handle turbulence with multi-scale vortices remains to be explored. We consider two additional directions for future work. First, we plan to explore the numerical accuracy of our neural vortex representation to improve the current vortex particle methods for scientific computing. Secondly, we plan to combine our differentiable simulator with neural rendering methods to synthesize visually appealing simulations from 3D videos.

ACKNOWLEDGMENTS

We thank all the anonymous reviewers for their constructive feedback. This work is in part supported by ONR MURI N00014-22-1-2740, NSF RI #2211258, #1919647, #2106733, #2144806, #2153560, the Stanford Institute for Human-Centered AI (HAI), Google, and Qualcomm.

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

## A    IMPLEMENTATION DETAILS

In this section, we describe the implementation details of our proposed method.

*Integrators.* As described above and illustrated in Figure 2, our system embeds two differentiable integrators in the loop. The Eulerian integrator is implemented using the Back and Forth Error Compensation and Correction (BFECC) (Kim et al., 2005) method for value look-up, and the 3$^{rd}$ order Runge-Kutta method for time-stepping. The Lagrangian integrator is implemented using the Forward Euler method.

*Network $N_1$.* The network $N_1$ adopts a series of 3 residue blocks with increasing widths $[64, 128, 256]$, whose architecture is similar to He et al. (2016) but with convolution layers replaced by linear layers with sine activation functions. The frequency factor $\omega_0$ discussed in Sitzmann et al. (2020) is set to 1.

*Network $N_2$.* The network $N_2(r, \delta)$ is structured as follows. First, the input $r$ is scaled by the input $\delta$ as $\bar{r} = r \cdot \frac{\eta}{\delta}$, where $\eta$ is a hyperparameter that corresponds to the characteristic scale of the vortices. Then, $\bar{r}$ is transformed into $\hat{r}$ as $\hat{r} = \bar{r}^{0.3}$, a reparametrization that stretches the value $\bar{r}$ near 0. This exploits the insight that the velocity varies more aggressively near a vortex. The value $\hat{r}$ is then fed through 4 residue blocks, which are the same as in $N_1$ but with a shared width of 40. The output from these residue blocks is scaled by multiplying with $\frac{\eta}{\delta}$. The scaled value is the output of $N_2(r, \delta)$, which is used for the velocity computation according to Equation 7.

*Training details.* Both the image loss and the velocity alignment loss are MSE, and the velocity alignment loss has an extra scaling factor of $0.001$. We use the Adam optimizer with $\beta_1 = 0.9$, $\beta_2 = 0.999$, and learning rates 0.0003, 0.001, 0.005, and 0.005 for $N_1$, $N_2$, $\Omega$ and $\Delta$ respectively. We use a step learning rate scheduler and set the learning rate to decay to $0.1$ of the original value at iteration 20000. We use a batch size of 4, so for each iteration, 4 starting times are picked uniformly randomly among $[0, 1, \ldots, t_E]$ for evaluation. The sliding-window size $m$ is set to 2.

*Pretraining $N_1$.* We pretrain $N_1$ for 10000 iterations with 2 objectives: (1) for all $t \in [0, t_E]$, $N_1(t) = [(\boldsymbol{x}_1)_t, \ldots, (\boldsymbol{x}_n)_t]$ coincide with the centers of a $4 \times 4$ grid, (2) for all $t \in [0, t_E]$, $\frac{dN_1}{dt} = \boldsymbol{0}$, so that these particles are initialized to be stationary. We use MSE for the positional and velocity losses, and the other training specifications are the same as described above.

*Computational performance.* Running on a laptop with Nvidia RTX 3070 Ti and Intel Core i7-12700H, our model takes around 0.4s per training iteration, and around 40000 iterations to converge (for a $256 \times 256$ video with 100 frames). For inference, each advance step costs around 0.035s.

## B    COMPARISON WITH DIFFERENTIABLE FLUID SIMULATION

We compare our method qualitatively and quantitatively against a standard, grid-based differentiable fluid simulator (referred to as Diff-Sim) on both synthetic and real videos. This baseline method is an auto-differentiable implementation of the method proposed by Fedkiw et al. (2001), which is a classic, widely-adopted numerical method for simulating vortical fluids. The method is designed to solve the 2D Euler equations for inviscid fluid, hence it can in theory recreate the inviscid fluid phenomena represented by any video if provided with the appropriate initial conditions and simulation parameters.

Therefore, in this experiment, we make use of its differentiable nature to optimize (1) the initial grid velocities (a $256 \times 256 \times 2$ tensor), and (2) the vorticity confinement strength, which is a scalar value, with the objective of minimizing the discrepancy between the simulated results and the input video. The loss computation between the simulated image sequence and the ground truth is the same as in our method. We note that the idea of optimizing initial conditions using differentiable fluid simulation to fit specific target frames has been explored in Hu et al. (2020). However, their task is notably simpler than ours, since they only require the simulated image to match a target frame at the end of the simulation, while our goal is to match the underlying motion of the entire video, and dynamically unroll into the future.

*Comparison on a synthetic video.* We start by comparing both methods on a synthetic video with 300 frames (the first 100 observed for training, the last 200 reserved for testing), which yields a visual comparison that can be found in Figure 10. We observe that our method successfully learns the dynamics represented in the video: the generated video and velocities closely resemble the ground

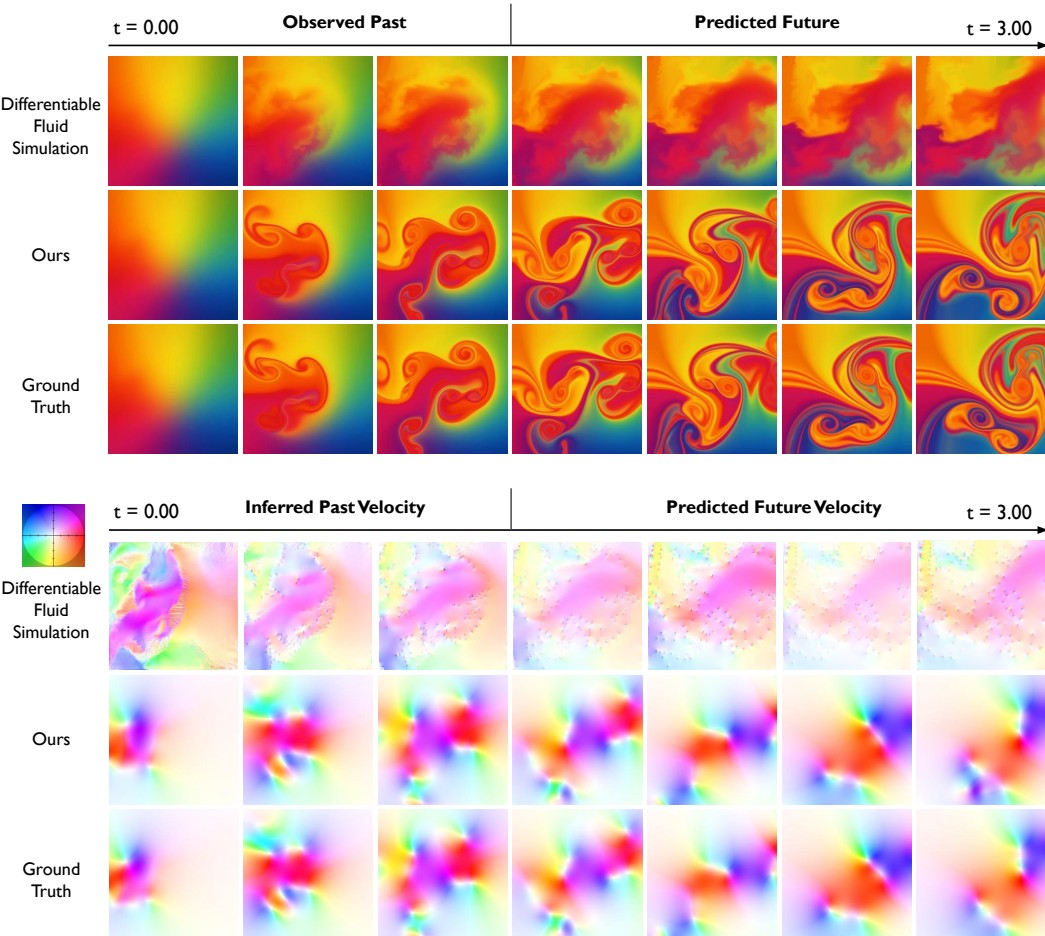

Figure 10: Visual comparison between a differentiable grid-based simulator and ours on a synthetic video. The upper half displays the simulated image, while the lower half displays the underlying velocity, whose color wheel is depicted. On the bottom row of each half is the ground truth sequence, which has 300 frames. The first 100 frames are available for both methods to learn from, while the rest are unseen during training.

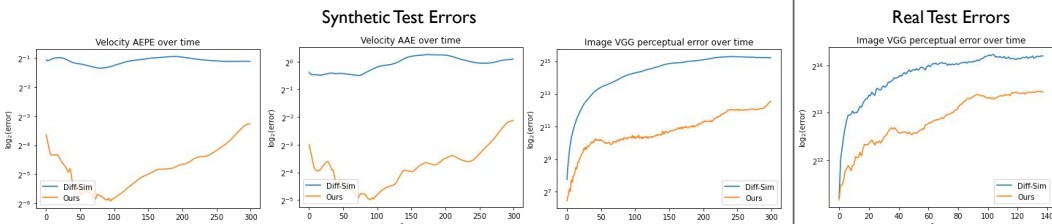

Figure 11: Error plots of the comparison between Diff-Sim and our DVP method on a synthetic dataset.

truth even in the unseen frames. Diff-Sim, on the other hand, shows a weak resemblance with the ground truth for the seen frames, but fails to capture the individual eddies in the video. Consequently, it fails to predict the future dynamics. Diff-Sim's lack of correspondence to the dynamics of the ground truth is also made evident in Figure 13. The result clearly suggests that our method has better learned the dynamical evolution. This performance discrepancy is numerically supported by the errors documented on the left panel of Table 3 and plotted on the left panel of Figure 11, both showing that our method yields reduced image-level and velocity-level errors compared to Diff-Sim.

*Comparison on a real video.* We then use the same experimental setup to perform learning on a real video with 139 frames (the first 93 observed for training, the last 46 reserved for testing), as depicted in Figure 12. We observe that on the real video, the same behavioral patterns for both

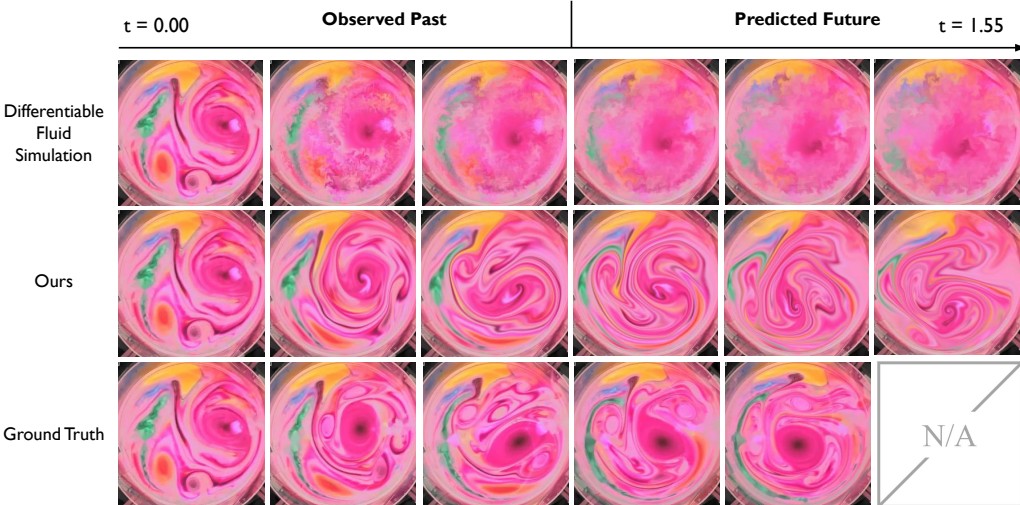

Figure 12: Comparison between Diff-Sim and our method on a real video.

| | Synthetic Video Errors | | | | | Real Video Errors | |
|---|---|---|---|---|---|---|---|
| | AEPE | AAE | Vort. | RMSE | VGG | VGG (avg.) | VGG (final) |
| Diff-Sim (Grid) | 0.469 | 0.953 | 24.43 | 0.157 | 26043 | 15076 | 18792 |
| Ours | **0.041** | **0.081** | **1.482** | **0.055** | **2171.4** | **7846.2** | **11081** |

Table 3: Error comparison between Diff-Sim and our method on synthetic and real videos.

systems on the synthetic one have carried over. For the results generated by Diff-Sim (top row), we can see that the overall, large-scale motion (the large eddy moving towards bottom-left) is faintly identifiable. Nevertheless, all the smaller vortices are missing, and the entire image quickly diffuses as the simulation proceeds. This can be attributed to the numerical diffusion issues innate to grid-based simulations, as well as the lack of embedded fluid structures. In comparison, our method well-preserves the vortical movements due to its built-in structure, and produces a plausible future rollout extending beyond the duration of the original video. Although both systems are unable to perfectly model the exact mechanism that governs this real-world video (due to unmodeled factors such as fluid viscosity, air friction, and 3-dimensional forces), our proposed method does a better job of retaining the vortical patterns and energetic flows thanks to its vorticity-based formulation and the Lagrangian-Eulerian design, as can be observed in the middle row of Figure 12. The advantage of our system over Diff-Sim on the real video is numerically supported, as shown on the right panels of Table 3 and Figure 11. Since we do not have the ground-truth velocities for real videos, we compare the VGG perceptual loss (Johnson et al., 2016) between the simulated sequence of both methods and the real video, which demonstrates quantitatively that our generated results better resemble the input video than those generated by the baseline.

## C  ADDITIONAL BENCHMARK TESTING

As depicted in Figure 14, to further illustrate our method's advantage and generalizability, we have conducted an additional set of numerical tests on another synthetic video (of 180 frames with the first 60 revealed for training), and compared our method's performance with 4 benchmarks in terms of both velocity inference quality and future prediction quality. The ground truth data is generated using a significantly different background image (sharp color tiles vs. smooth color gradients), and a different velocity kernel (second-order Gaussian kernel vs. first-order Gaussian kernel) (Beale & Majda, 1985). The experimental setup is otherwise the same as the one presented in the main text (in Figure 7), with the same compared benchmarks.

The comparison of the velocity inference quality can be found in Figure 16 and the top panel of Figure 15. Figure 16 depicts the uncovered velocities of frame 40 (among the 60 input frames)

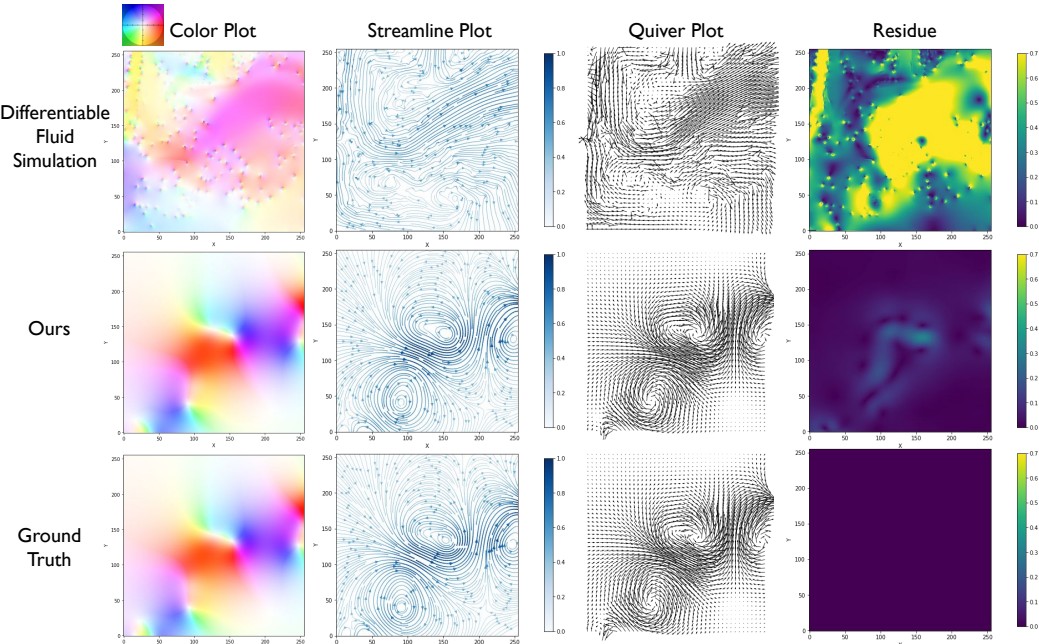

Figure 13: Comparing the quality of the velocity predicted by our method and Diff-Sim. We show the predicted velocity (of frame 200) in three different forms (color, streamline, and quiver plots) in addition to the residue (end-point error) compared to the ground truth velocity.

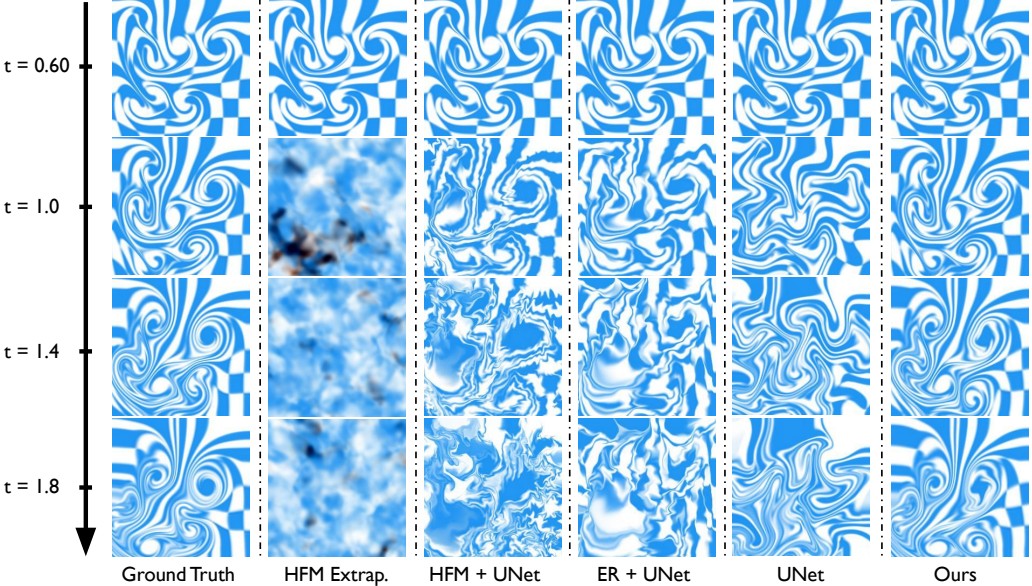

Figure 14: Future prediction results: our method compared to the baselines on a synthetic video.

by all 4 methods compared to the ground truth. The top row depicts the respective velocities in colors with the color wheel supplied; the middle row depicts the velocities in streamlines; and the bottom row depicts the velocity residues compared to the ground truth, measured in end-point error (EPE). As with the results in Figure 5, we can see that HFM, UNet, and our method can all infer the underlying velocity field at high precision, whereas E-R yields a visibly noisier approximation. As seen on the bottom row, the inference performance of UNet and Ours are very close, but our method takes the slight edge with an average error (AEPE) of 0.0143, which is 33.49% less than the error of 0.0215 yielded by UNet. The advantage of our method is not unique to the specific frame selected. As plotted on the top row of Figure 15, it can be seen that our method (red) consistently

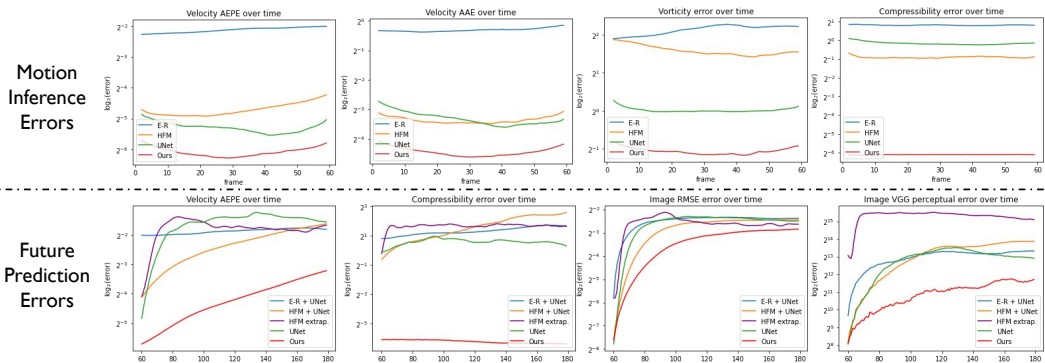

Figure 15: Error analysis on a second synthetic video. The top row plots the inference errors of velocity, vorticity, and compressibility. The bottom row plots the future prediction errors, which consider both the dynamic error of the velocity and the perceptual error of the generated image sequence.

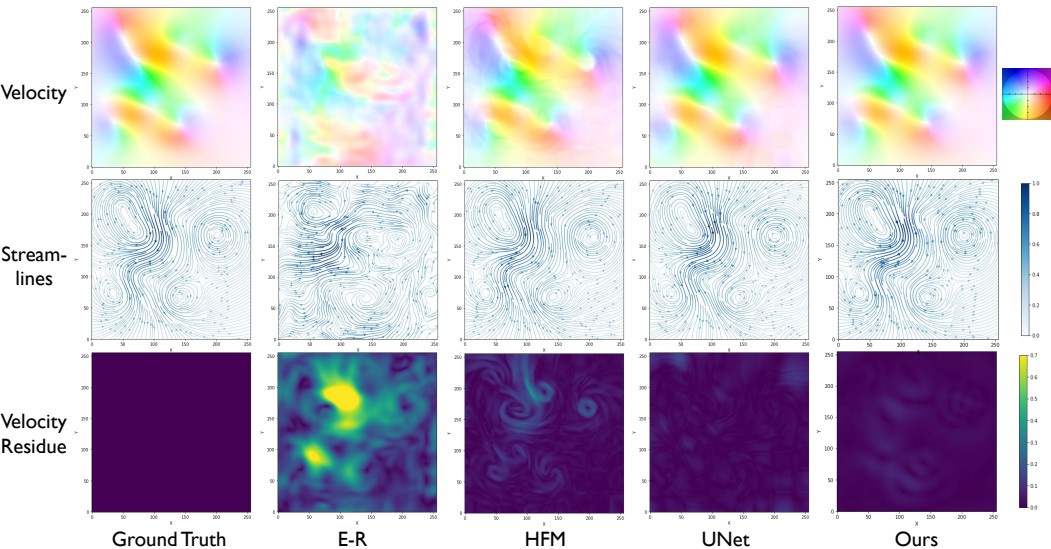

Figure 16: Baseline comparison of velocity inference on a synthetic video. Our method recovers the underlying velocity field with the highest accuracy.

yields the lowest velocity-inference error throughout the 60 input frames, in terms of the average end-point error (AEPE), average angular error (AAE), vorticity RMSE and compressibility RMSE. The time-averaged errors of these metrics are documented in Table 4, which again shows that our method yields the best estimations.

*Future prediction.* We also compare our method's future prediction results with the baselines. In Figure 14, we show a visual comparison of all 5 methods against the ground truth. It highlights the close resemblance of our generated sequence with the ground truth, which is twice as long as the sequence used for training. Compared to the baselines, our method yields the best match to the ground truth video, capturing the accurate vortical flow structures. HFM+UNet, E-R+UNet, and UNet can generate reasonable future predictions up to $t = 1.0$ (for 40 frames). For $t > 1.0$, these sequences start to distort in different ways, due to their lack of physical structures and constraints. The direct extrapolation of HFM yields the least plausible results, quickly degrading to noise. We compare these sequences quantitatively using the 4 velocity-based metrics, along with the 2 image-based metrics: the pixel-level RMSE and the VGG feature reconstruction loss. Four of these time-dependent errors are plotted in the bottom row of Figure 15, with their time-averaged counterparts documented on the right of Table 4. In summary, we observe that our method outperforms the existing baselines for this video both quantitatively and qualitatively.

| | Time-averaged Inference Errors | | | | | Time-averaged Prediction Errors | | | | | |
|---|---|---|---|---|---|---|---|---|---|---|---|
| | AEPE | AAE | Vort. | Div. | | VGG | RMSE | AEPE | AAE | Vort. | Div. |
| E-R | 0.229 | 0.805 | 4.380 | 1.750 | +UNet | 8138.8 | 0.178 | 0.272 | 1.115 | 4.694 | 2.504 |
| HFM | 0.038 | 0.097 | 3.001 | 0.533 | +UNet | 9389.5 | 0.146 | 0.201 | 0.715 | 10.58 | 3.199 |
| | | | | | Extp. | 40967 | 0.166 | 0.293 | 1.221 | 4.862 | 3.152 |
| UNet | 0.026 | 0.101 | 1.013 | 0.895 | | 7721.3 | 0.170 | 0.330 | 1.141 | 5.462 | 1.496 |
| Ours | **0.015** | **0.046** | **0.480** | **0.015** | | **2045.0** | **0.097** | **0.057** | **0.173** | **1.547** | **0.013** |

Table 4: Time-averaged errors of our method compared to various baselines on a synthetic video.

## D    NUMBER OF VORTEX PARTICLES

In our proposed method, we use $n$ vortex particles to learn fluid dynamics. However, we note that *vortices* are not intrinsic to fluid phenomena, but are rather imposed constructs to allow fluids to be better understood conceptually and modeled numerically. Thus, the number of vortices $n$ is fundamentally a hyperparameter that does not admit a uniquely-correct value.

With this in mind, we let $\hat{n}$ denote the minimum number of particles that can be used to model the fluid system to an acceptable accuracy. This natural number $\hat{n}$ surely exists since it has been proven that vortex particle methods converge to the exact solution of the 2D Euler Equations (Beale & Majda, 1985; Hald, 1979). We are mostly concerned with the cases where $n > \hat{n}$, which means the number of deployed degrees of freedom (DoFs) is greater than the number necessary for the given fluid system. In the following section, we show that our method can spontaneously prune the redundant vortices and thus it is robust to a reasonable range of $n > \hat{n}$. In Figure 17, we show the results of learning the same underlying motion with 4, 9, and 64 vortex particles. In Figure 18, we show the underlying velocity and vorticity fields using different numbers of vortex particles.

*Spontaneous pruning of redundant DoFs.* As shown on the top row of Figure 17, the ground truth is generated with 4 vortices, so it is safe to assume that $\hat{n} = 4$. Learning with 4 vortices (as shown on the second row) represents the case where $n = \hat{n}$. Comparing the first row with the second row, we can see that there is a one-to-one correspondence between the ground-truth vortices and the learned vortices, with each learned vortex assuming the role of one individual ground-truth vortex (obtaining the same vorticity and initial position).

When we have 9 vortices (third row), there are more vortex particles than those in the ground truth. In this case, two interesting phenomena occur to spontaneously prune these redundant particles: degeneration and clustering. First, some particles degenerate themselves by reducing their strengths to 0 or by moving farther away from the domain. We can observe both mechanisms taking place on the two lingering particles on the top part of the third row. They both have low strengths (evident from their turquoise color) and are peripheral to the domain. Secondly, multiple particles can aggregate to emulate a single particle with greater strength. Since the velocity computation is a distance-weighted summation (as in Equation 7), if multiple particles coincide at the same location, they effectively act as one single particle with their vorticities added together. This phenomenon can be observed by comparing the lower halves of the second and third rows. Both of these mechanisms enable our system to spontaneously prune redundant vortices. In the last row, we show that our method is robust to even 64 vortices.

Figure 19 helps to illustrate this spontaneous pruning mechanism by showing different snapshots of the training process. Shown on the left are the vortex particles' behaviors soon after the training has begun. It is particularly noticeable that, on the bottom row, the 64 particles are scattered throughout the fluid domain, and the simulated result appears quite different from the ground truth. Moving from left to right, these particles become more and more clustered around the flow regions, with much fewer "freelance" particles, and the end result can approximate the ground truth much better.

Finally, we note that $n < \hat{n}$ is still challenging to resolve as the system would be over-constrained. Nevertheless, we empirically find that $n = 16$ is sufficient for all the real and synthetic videos we consider in our experiments.

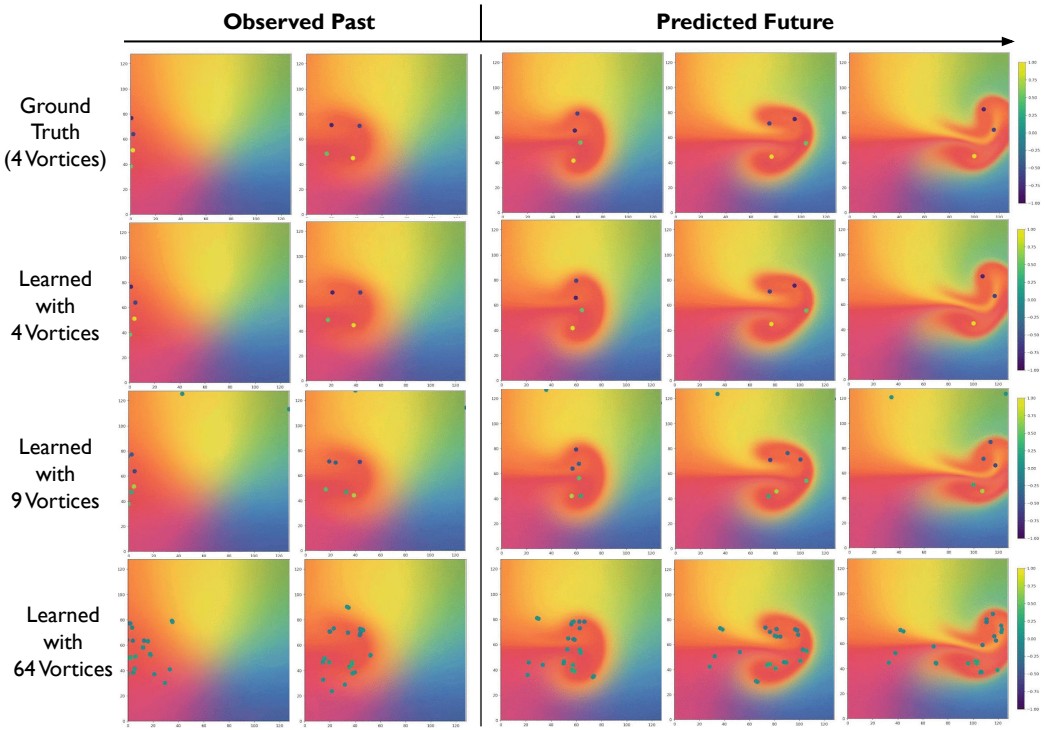

Figure 17: The same underlying motion learned with different numbers of vortex particles. The ground truth has 100 frames; the first 30 frames are observed during training, and the remaining 70 frames are predicted.

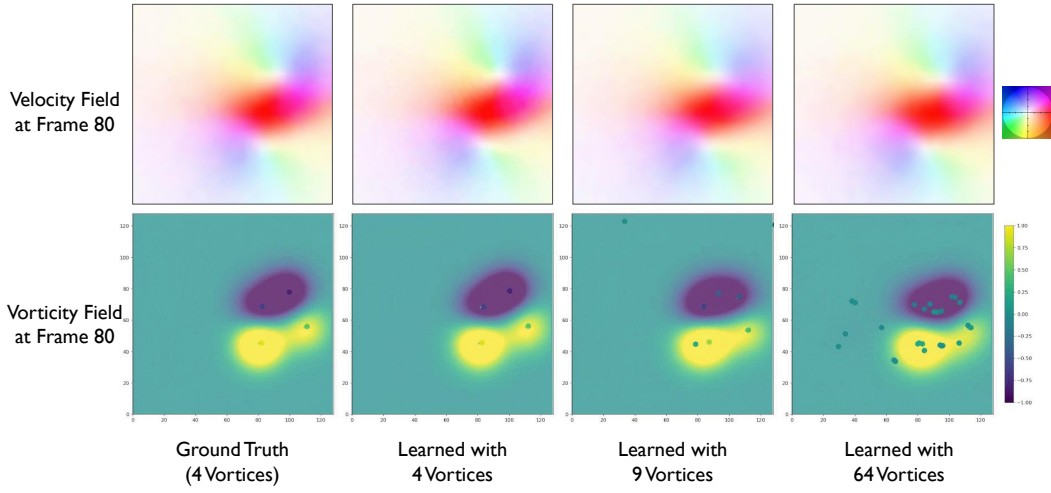

Figure 18: Robustness of our DVP method against different numbers of vortices used. Different numbers of vortices can represent similar underlying dynamics.

# E ABLATION: LEARNABLE VELOCITY KERNEL

In traditional vortex simulation applications in Computer Graphics and Computational Fluid Dynamics, the velocity kernel is hand-selected (typically from Gaussian kernels of different orders) with a uniform support radius (size). Such approaches are designed to perform forward simulation, yet they are limiting when used for backward inference tasks, *i.e.*, to reconstruct input videos. In our method, we address this issue by learning neural kernels with learnable sizes. By leveraging data-

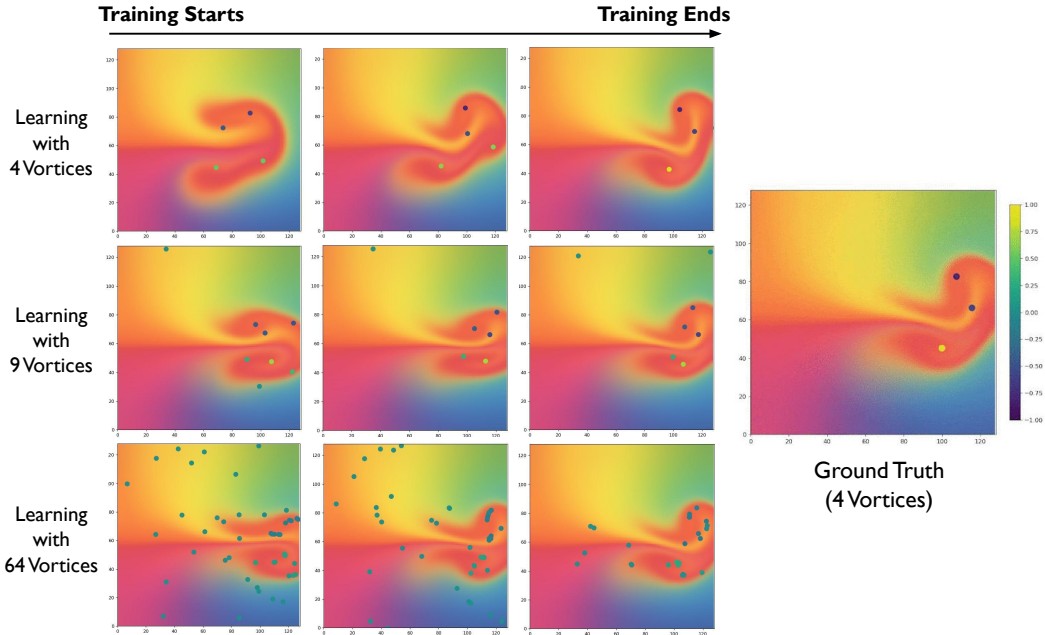

Figure 19: The training evolution when using different numbers of vortex particles.

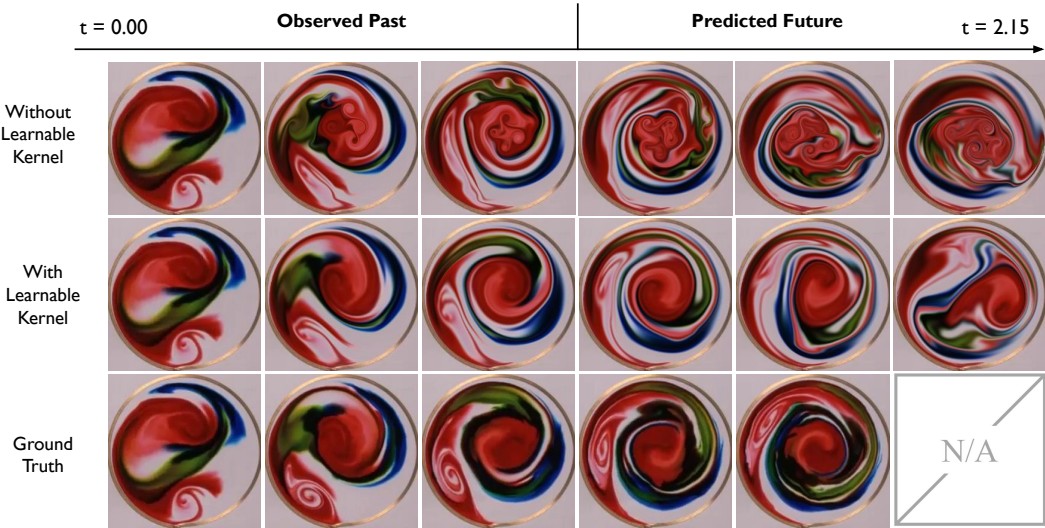

Figure 20: Ablation study: reconstruction and prediction on a real video with learnable velocity kernels (our full method) and without learnable velocity kernels (ablation).

driven techniques, we can reconstruct and predict fluid flows that not only are visually pleasing, but also resemble the specific dynamical traits embodied in the input video.

In Figure 20, we present an ablation study on the learnable velocity kernels. We reconstruct and predict a real-world video using our method and an ablated version in which the learnable kernel is replaced with a hard-coded first-order Gaussian kernel with uniform size. The ground truth, shown on the bottom row, has 126 frames revealed for training and 62 frames hidden for testing. In the middle row, we learn to fit the video with our learnable kernel enabled. In the top row, we learn to do the same with the learnable kernel disabled. It can be observed that the middle row well-captures the characteristic smoothness of the flow, and simulates an image sequence that resembles the ground truth. The ablated version (top row) can also learn the correct overall motion (clockwise rotation), but it induces various smaller eddies and wrinkles uncharacteristic of the input video.

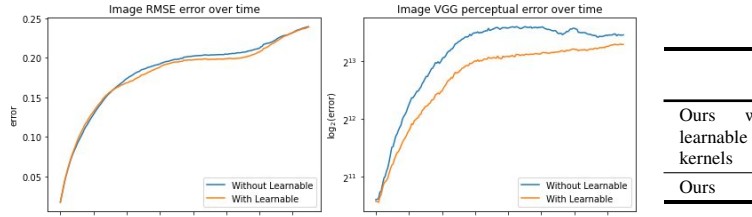

| | VGG (avg.) | VGG (final) | RMSE (avg) |
|---|---|---|---|
| Ours w/o learnable kernels | 9698.7 | 11240 | 0.183 |
| Ours | **7365.6** | **10027** | **0.180** |

Figure 21: Time-varying losses corresponding to Figure 20. Table 5: Time-averaged errors of Figure 21.

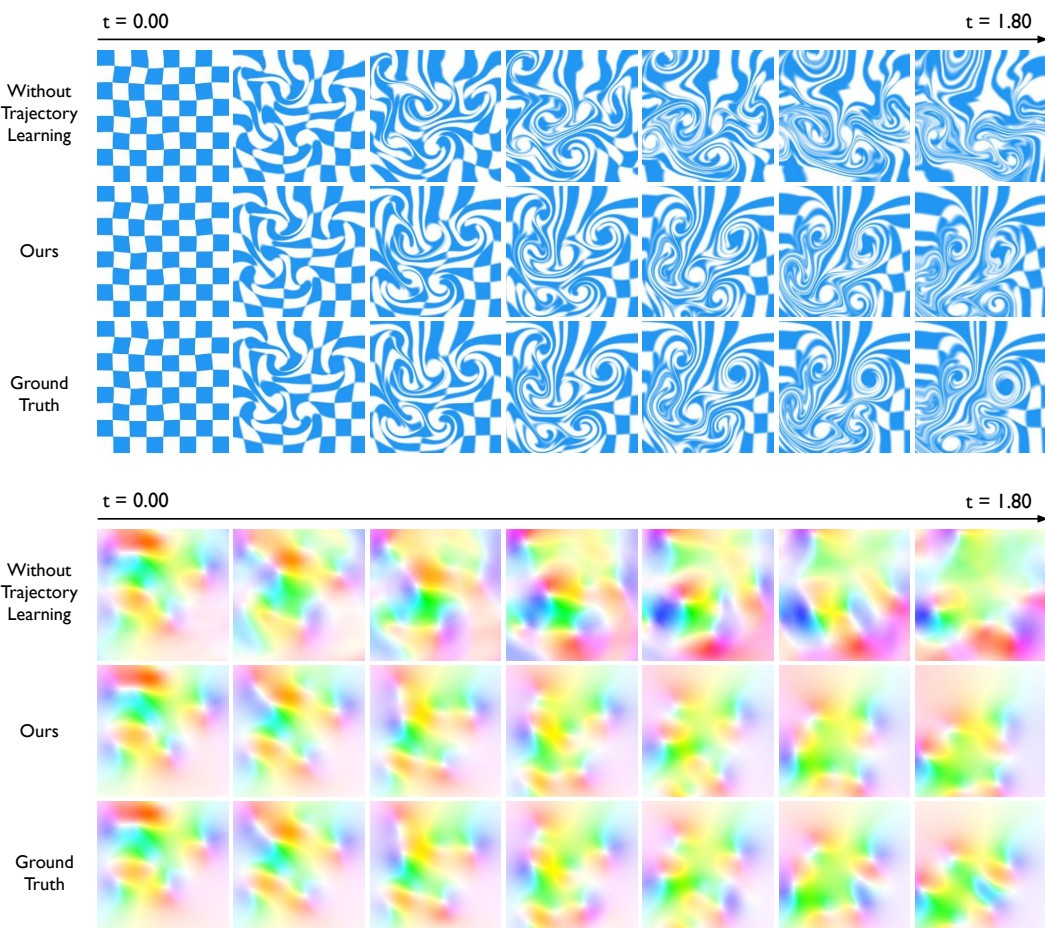

Figure 22: We compare our method against its ablated version which does not have the feature trajectory learning. The top depicts the simulated images, and the bottom depicts the simulated velocities. The results of both approaches are compared to the ground truth.

Extending to unseen frames, our method can continue to retain the overall structure of the eddies, while the ablated version (without the learnable kernel) drives the pattern to disintegrate, and develops various folds and wrinkles that do not resemble the dynamical characteristics of the real video. We further show quantitative results plotted in Figure 21 and documented in Table 5. In summary, learning the velocity kernels allows for better reconstruction and prediction of fluid flow specific to the input video.

## F ABLATION: TRAJECTORY LEARNING

In our approach, we learn the full trajectories of vortex particles for the input video. An alternative is to learn the initial condition only. However, we find that the former option is more computationally tractable and effective, since it can exploit the full range of the input video at a manageable cost. To see this, suppose we have 100 training frames in the video, and the goal is to infer the initial

| | Velocity Errors | | | | Image Errors | |
|---|---|---|---|---|---|---|
| | AEPE | AAE | Vort. | Div. | RMSE | VGG |
| Ours (Ablated) | 0.257 | 0.753 | 4.689 | 0.054 | 0.170 | 6759.4 |
| Ours | **0.043** | **0.131** | **1.180** | **0.014** | **0.081** | **1805.9** |

Table 6: Time-averaged velocity-level and image-level errors by our method and its ablated version without trajectory learning.

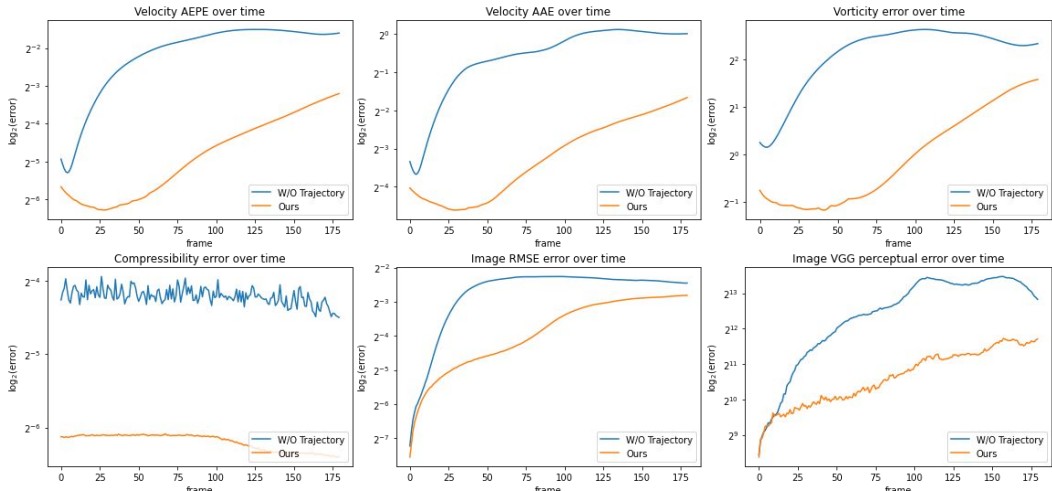

Figure 23: Time-varying velocity-level and image-level errors by our method and its ablated version without trajectory learning.

condition at frame-1. If we directly optimize the initial condition using the last frame, we need to simulate from frame-1 all the way to frame-100, compute the loss and backpropagate. Unrolling such a long sequence for each training iteration (1) takes a long time, (2) leads to noisy gradients, and (3) is practically infeasible due to memory constraints. On the other hand, learning the whole trajectory allows us to address these challenges by using a smaller sliding window in time (*e.g.*, simulating only 3 frames at a time) and aggregating the dynamics information throughout the whole video. In Figure 22, we show a comparison of both methods in action, with a total of 180 simulated frames. On the top panel, we show the reconstruction and prediction results for both our full method and an ablated version where we directly learn the initial condition. Note that the ablated version can only unroll the first 13 frames (and thus is learned using only the input video's first 14 frames) due to the memory constraint (which is consistent for both candidates). In contrast, our full method can handle the 60 input frames like in the setup of Appendix C. On the bottom panel, we show the velocity corresponding to the top panel. We observe that our method and its ablated version can approximate the ground truth reasonably well at the beginning of the simulation (the left three images). However, the ablated version starts to distort significantly in terms of both the advected image and the underlying velocity. This observation is in agreement with the numerical evidence, as plotted in Figure 23 and documented in Table 6, which shows that our full method consistently outperforms its ablated counterpart across all metrics. We conjecture that the underlying reasons for this performance discrepancy are threefold: first, the ablated version can only learn from the beginning section of the fluid observation, which provides limited information to correctly infer the initial condition. Secondly, only learning the initial condition is more susceptible to accumulated errors than our full method. Thirdly, using a limited number of frames makes it harder to synthesize an appropriate velocity kernel. In summary, our observations suggest that learning the full trajectory is more desirable than learning the initial condition only.

