# OpenReview forum: "Learning Vortex Dynamics for Fluid Inference and Prediction"
_ICLR.cc/2023/Conference — ICLR 2023 poster_

### Official Review · Reviewer_jKRP · 2022-10-23

**Confidence:** 3
**Clarity, Quality, Novelty And Reproducibility:** See above.
**Correctness:** 3
**Technical Novelty And Significance:** 3
**Empirical Novelty And Significance:** 3
**Recommendation:** 6

**Strength And Weaknesses:**

Strength:
1. Instead of modeling the entire image space dynamics, the method models only the vortex information, which effectively reduces the dimension and difficulty, and also helps generalize to unseen cases.
2. The results are very promising. They clearly outstand previous works.
3. The experimental comparisons are thorough and complete. They support the claims well.

Weakness:
1. There should be more detailed descriptions in the method section. For example, how are the vortex positions initialized? Are they inferred from the image or overfitted to the given example? How does the model determine how many vortices to model given an image? I don't see a module to encode this information from the input image to the vortex space.
2. The strengths and the sizes of all vortices are assumed to be the same throughout the entire prediction. Is it mostly the case in the real world? How does it introduce limitations to this model? Again, how to model the initial values using trainable vectors is unclear. Do they overfit to a certain example? If not, how are they estimated from the image?
3. I wonder if this task can be achieved already by optimizing the parameters using differentiable fluid simulation:
Takahashi, Tetsuya, et al. "Differentiable fluids with solid coupling for learning and control." Proceedings of the AAAI Conference on Artificial Intelligence. Vol. 35. No. 7. 2021.
Please add some discussions regarding the difference between the proposed method and the differentiable simulation system.

**Summary Of The Paper:**

This paper introduces a learning-based method to predict vortex dynamics. It designs a vortex dynamics module to learn the evolution of the vortices and another network to transform from the vortex space to the image space velocity field. Results show that the proposed method provides a realistic and stable vortex evolution trajectory, both in synthetic and real-world videos.

**Summary Of The Review:**

Although the method and the results are mostly solid, there are some clarity issues regarding the method itself which can prevent the readers from understanding the whole picture. I would recommend weak reject now, but will be happy to change once the issues are addressed.

---

> ### Author Response · Authors · 2022-11-19
> **Response to Reviewer jKRP**
>
> Thank you very much for your constructive and insightful review. Please find our response below:
>
> **Q1**: There should be more detailed descriptions in the method section. For example, how are the vortex positions initialized? Are they inferred from the image or overfitted to the given example? How does the model determine how many vortices to model given an image?
>
> **A1**: Thanks for the suggestion. For all of our experiments, we initialize 16 vortices regularly spaced on the cell centers of a 4 by 4 grid. In practice, this is done by pretraining the trajectory network $N_1$ so that its evaluation at time 0 coincides with the grid points. In the updated manuscript, this initialization scheme is discussed in the added “*Trajectory Initialization*” paragraph on Page 4, and the details regarding pretraining is given in Appendix A. Furthermore, a discussion with experiments on the number $n$ of deployed vortices is given in Appendix D. In short, we deploy a superfluous number of vortex particles, and let the system spontaneously prune the unused degrees of freedom through clustering or degeneration.
>
> These vortex positions are overfitted to a particular input video, as our goal is to infer and predict fluid dynamics from a single RGB video without any additional training data.
>
> **Q2**: The strengths and the sizes of all vortices are assumed to be the same throughout the entire prediction. Is it mostly the case in the real world? How does it introduce limitations to this model? Again, how to model the initial values using trainable vectors is unclear. Do they overfit to a certain example? If not, how are they estimated from the image?
>
> **A2**: The time-invariance of the strengths and sizes are valid assumptions for the branch of fluid problems that we tackle: 2-dimensional, inviscid fluids with conservative body forces. In these fluid systems, Kelvin’s circulation theorem holds, which suggests that the integral of vorticity $\omega$ over the control area (i.e., the vortex strength) of a particle moving with the flow is conserved, and the control area (i.e., the vortex size) of the particle remains constant due to fluid incompressibility (please refer to [Hald, 1979] for further details). We have also added this argument to the first paragraph of Section 4.1.
>
> This time-invariance assumption should be abandoned if we wish to generalize to 3D and/or viscous flow. Advantageously, dealing with the added vortex stretching term and the viscous diffusion term boils down to evolving the vortices’ strengths and sizes, as have been detailed in section 2 and 3 of [Mimeau and Mortazavi. 2021]. In other words, replacing the time-invariant strengths and sizes by evolving ones, with the evolution represented by a learnable neural network, is in theory the sole missing component before our method gets extended to general, 3D viscous fluids. We have included this insight in the Section 6 (conclusion) to encourage future explorations.
>
> The trainable vectors ($\Delta$ for sizes and $\Omega$ for strengths) overfit to the input video. Initially, they are all set to be 0. So the particle sizes are all $0.5 + \epsilon$ (a small value added to prevent singularity), and the particle strengths are all 0, which correspond to the activation functions mentioned in the "*Learning Particle Trajectory*" paragraph (page 4 in our main paper). Then, by iteratively running differentiable vortex simulation with the current strengths and sizes and comparing the result with the input video, the $\Delta$ and $\Omega$ are iteratively updated through gradient descent to better fit the video frames.
>
> We invite our reviewer to revisit Section 4.1 (Differentiable Vortex Particles), which we have carefully revised to improve the expositional clarity.

---

> > ### Comment · Reviewer_jKRP · 2022-11-22
> > **Thanks**
> >
> > Thank you for the responses. My concerns are all well addressed. I would like to increase my score to weak accept.

---

> ### Author Response · Authors · 2022-11-19
> **Response to Reviewer jKRP (continued)**
>
> **Q3**: I wonder if this task can be achieved already by optimizing the parameters using differentiable fluid simulation: Takahashi, Tetsuya, et al. "Differentiable fluids with solid coupling for learning and control." Proceedings of the AAAI Conference on Artificial Intelligence. Vol. 35. No. 7. 2021. Please add some discussions regarding the difference between the proposed method and the differentiable simulation system.
>
> **A3**: This is a good question. Since Takahashi et.al. [2021] does not provide open-source implementation, we implement a classic fluid solver [Fedkiw, Stam and Jensen, 2001], using PyTorch to make it differentiable. This is a reasonable baseline since a similar differentiable implementation of this method has been demonstrated by Hu et.al. [2020] to successfully fit a given **target image**. However, in Appendix B, we show that such a differentiable simulator **cannot** successfully fit our **target videos** or predict future dynamics.
>
> To train a differentiable fluid simulator to fit a given fluid video, the parameters that need to be trained are 1) the initial condition of the fluid system (e.g., initial velocity) and 2) the tunable parameters used during the numerical simulation. Thus, we use the differentiable fluid simulator to learn the initial condition and the simulation parameters so that the simulated motion matches that of the video. We conduct the experiment on both real and synthetic videos. For both cases, we observe that the grid-based differentiable simulation can only faintly capture the large-scale movements, and yet the structures of the eddies are notably inaccurate, due to the lack of physical structure in its representation. In comparison, our method learns the underlying fluid dynamics more successfully, from both the visual and the numerical standpoint. More details can be found in the added Appendix B in our revised manuscript. We also note that for the differentiable simulator baseline, each training iteration needs to run 100 steps (given a video with 100 frames), and each step needs over 100 Jacobi iterations for the pressure solve, making it train much slower than our proposed method (35s per iteration vs 0.4s per iteration).
>
> **References:**
>
> [Hald, 1979] Hald, Ole H. “Convergence of Vortex Methods for Euler’s Equations. II.” SIAM Journal on Numerical Analysis 16, no. 5 (1979): 726–55.
>
> [Beale and Majda, 1982] Thomas Beale and Andrew Majda. "Vortex methods. II. Higher order accuracy in two and three dimensions." Mathematics of Computation 39.159 (1982): 29-52.
>
> [Mimeau and Mortazavi, 2021] Chloé Mimeau and Iraj Mortazavi. "A review of vortex methods and their applications: From creation to recent advances." Fluids 6.2 (2021): 68.
>
> [Fedkiw, Stam and Jensen, 2001] Ronald Fedkiw, Jos Stam, and Henrik Wann Jensen. "Visual simulation of smoke." SIGGRAPH, 2001.
>
> [Hu et.al., 2020] Yuanming Hu, Luke Anderson, Tzu-Mao Li, Qi Sun, Nathan Carr, Jonathan Ragan-Kelley, and Frédo Durand. "Difftaichi: Differentiable programming for physical simulation." ICLR, 2020

---

### Official Review · Reviewer_PYMt · 2022-10-23

**Confidence:** 3
**Correctness:** 4
**Technical Novelty And Significance:** 3
**Empirical Novelty And Significance:** 3
**Recommendation:** 8

**Clarity, Quality, Novelty And Reproducibility:**

The paper is clearly written and well organized. But some implementation details are missing such as the number of particles n and the initialization of $\Delta$ and $\Omega$. Given my limited knowledge in this field, this work looks novel to me and the proposed method would be useful to the community.

**Strength And Weaknesses:**

Strength:

- The proposed method of using a Lagrangian vortex particle representation and a learnable vortex-to-velocity mapping looks pretty interesting and reasonable to me. The vortex particle representation is more compact and reduces the problem's dimension. The learnable vortex-to-velocity mapping well connects the vortex particle and the fluid phenomena, making the pipeline end-to-end trainable. The model design also integrates fluid simulation and learning-based approach in a reasonable way, combining the advantages of both fields.

- The advantage of the proposed method over baselines is obvious in experiments. Quantitative and qualitative study in synthetic and real data demonstrate that the proposed method estimates more accurate velocity and predicts more plausible future sequences.

Weaknesses:

- The description of implementation is not detailed enough as some hyperparameters are missing. For example, the number of particles n is not provided. It is also not mentioned how $\Delta$ and $\Omega$ are initialized. Besides, it is suggested to also mention the efficiency of the method, like the training and inference time.

- The amount of data used for evaluation seems limited. If I understand correctly, the quantitative study is conducted on only one synthetic video and one real video, which only covers limited diversity. It would be better to evaluate on more videos to strengthen the results.

- The method now assumes inviscid fluid and 2-dimension, which is a somewhat constrained setting and would limit the application scope.

**Summary Of The Paper:**

This paper proposes a method to learn vortex dynamics from a single fluid video and thus be able to infer and predict fluid dynamics. At the core of the method are a Lagrangian vortex particle representation and a learnable vortex-to-velocity dynamics mapping. These two components can be trained merely from the observed video in an end-to-end manner. After training, the hidden physics quantities such as velocity field would be obtained, and the prediction of future states is also supported. The proposed method is compared with existing methods on both synthetic and real videos, where the proposed method shows more accurate velocity estimation and more plausible future prediction.

**Summary Of The Review:**

I am not familiar with all the literature in this field, but to the best of my knowledge, this paper looks pretty interesting to me. It cleverly combines Lagrangian particle representation and Eular representation in a unified and end-to-end trainable framework, addressing several existing challenges. The parameterization of position x and the vortex-to-velocity mapping using neural networks is reasonable. But more implementation details could be added and the evaluation can be improved by using more videos. In general, I am in favor of accepting this paper. But it is possible that I may overlook some issues given my limited knowledge in this field, so I am also interested to see other reviewers' comments.

---

> ### Author Response · Authors · 2022-11-19
> **Response to Reviewer PYMt**
>
> Thank you very much for your constructive and insightful review. Please find our response below:
>
> **Q1**: The description of implementation is not detailed enough (the number of particles $n$, how $\Delta$ and $\Omega$ are initialized, etc.)
>
> **A1**: Thanks for the suggestion. We have provided these details in our revised manuscript. Especially, we have dedicated Appendix A for a number of additional details to be found. Regarding the number $n$ of particles, we have also carried out one additional experiment (Appendix D) to show that the exact number of particles does not affect the performance as long as there are sufficient ones, because the redundant particles would spontaneously cluster or degenerate, effectively pruning the superfluous degrees of freedom. The initialization of $\Delta$ and $\Omega$ are described and discussed in the added “*Trajectory Initialization*” paragraph on Page 4.
>
> **Q2**: Mention the efficiency of the method, like the training and inference time.
>
> **A2**: Thanks for the suggestion. We have included these efficiency details as part of the added Appendix A.
>
> **Q3**: It would be better to evaluate on more videos to strengthen the results.
>
> **A3**: Thanks! We have included additional numerical and qualitative evaluations on additional video sequences in our revised paper. In Appendix C, we’ve carried out an additional benchmark testing against 4 baselines on a new synthetic video generated using a different image pattern and dynamics kernel. Furthermore, in Appendix B, we compare our method against a standard grid-based differentiable simulator on one synthetic and one real video (different from the one already used for quantitative studies). In Appendix D, we analyze our system’s response to different choices of the hyperparameter $n$ using a new synthetic video. In Appendix E, we showcase qualitatively and quantitatively the importance of data-driven velocity kernels on a real video. Finally, in Appendix F we test to verify the importance of learning the trajectory. In this way, quantitative studies are performed on 3 real videos and 2 synthetic videos, and qualitative studies are performed on another 1 synthetic video.
>
> **Q4**: The method now assumes inviscid fluid and 2-dimension, which is a somewhat constrained setting and would limit the application scope.
>
> **A4**: This is a good point, although we wish to note that the formalism we introduce is generalizable to learning viscous, 3-dimensional flows. The technical gap here would be the treatment of vortex stretching and viscous diffusion. In the literature of vortex methods in computational fluid dynamics, both of these phenomena can be handled by modifying the vortex strengths and sizes (please see section 2 and section 3 of [Mimeau and Mortazavi, 2021] for more detail). Therefore, a reasonable extension to our current method would simply be allowing the time-invariant tensors: $\Delta$ for sizes and $\Omega$ for strengths, to evolve with time, and use an additional neural network to approximate their evolution.
>
> **References:**
>
> [Mimeau and Mortazavi, 2021] Chloé Mimeau and Iraj Mortazavi. "A review of vortex methods and their applications: From creation to recent advances." Fluids 6.2 (2021): 68.

---

### Official Review · Reviewer_PvPD · 2022-10-24

**Confidence:** 3
**Correctness:** 4
**Technical Novelty And Significance:** 4
**Empirical Novelty And Significance:** 4
**Recommendation:** 8

**Clarity, Quality, Novelty And Reproducibility:**

Clean formulation and is very clearly presented. The formulation is non-trivial -- it is a good example of using insights from physics to solve complex learning tasks. The approach is quite specialized, but I do not see that as a problem. Very novel unless I am aware of similar concepts having been used in physics simulations.

Would be great if the authors provide code. Should be reproducible by someone with the necessary physics background.

**Strength And Weaknesses:**

+ Elegant and well-grounded formulation
+ Compelling results on both synthetic and real sequences
+ Section 3 and Figures 2/3 nicely explain the key ideas behind the approach

- Limited to 2D in its current version
- Are there invariants in vortex-based formulation that can be incorporated?
- Other limitations are discussed at the end of the paper but they should encourage future works

**Summary Of The Paper:**

I found this paper to be very interesting. The idea of having two 'simulators' one for vorticity for physics tracking and one for RGB video tracking is clever, and the coupling between vorticity/velocity in Equation (3) is something I was unfamiliar with. Although limited to 2D, in its current formulation, the idea of being able to link a visible and hidden simulator is intriguing. The results on both synthetic and real video captures are compelling and clearly perform better than other methods compared against.

**Summary Of The Review:**

An elegant solution to a difficult problem. Compelling demonstration on real video sequences. Not often such a coupling can be obtained between visible and hidden states to regularize a solution. The hidden states have the right complexity to regularize the problem!

---

> ### Author Response · Authors · 2022-11-19
> **Response to Reviewer PvPD**
>
> Thank you very much for your constructive and insightful review. Please find our response below:
>
> **Q1**: Limited to 2D in its current version.
>
> **A1**: Despite our method being limited to 2D inviscid flows in the current version, we’d like to point out that this formalism provides an advantageous framework to generalize to 3D viscous flows. To do this, two additional factors need to be modeled: vortex stretching and viscosity. To deal with vortex stretching, we may modify the vortices’ strengths based on the velocity field as they move. To deal with viscosity, we may either change the vortices’ strengths (Particle Strength Exchange Method) or sizes (Core-Spreading Method) as they move (please see [Mimeau and Mortazavi, 2021] for more details). Hence, the only extension needed is to change the sizes $\Delta$ and strengths $\Omega$ from time-invariant quantities to evolving quantities, and learn their interaction using another neural network. We envision that extending our framework to 3D would allow for further opportunities in modeling more general and complex real fluid videos.
>
> **Q2**: Are there invariants in vortex-based formulation that can be incorporated?
>
> **A2**: Yes. In fact, a large appeal of vortex-based methods compared to grid-based methods is the built-in conservation of fluid invariants (circulation, linear and angular impulses). The simplicity of our current approach already exploits the conservation of circulation (Kelvin’s circulation theorem) and the invariance of density (incompressibility), which together allow us to assume that a vortex particle’s strength and size (control area) are unchanged throughout simulation (please see [Hald, 1979] for more). In 3D scenarios, it is also proven that the vorticity is carried by a Lagrangian element and deformed by the Jacobian of the flow map (see Proposition 2 of [Cortez, 1995]). It can be said that the vorticity-based formulation leaves much room for physically-based designs to embed fluid invariants.
>
> **Q3**: Other limitations are discussed at the end of the paper but they should encourage future works.
>
> **A3**: Thanks! We have updated our conclusion section with the following text to encourage future explorations:
>
> “In this work, we propose a novel data-driven system to perform fluid hidden dynamics inference and future prediction from single RGB videos, leveraging a novel, vortex latent space. The success of our method in synthetic and real data, both qualitatively and quantitatively, suggests the potential for embedding Lagrangian structures for fluid learning. Our method has several limitations. First, our vortex model is currently limited to 2D inviscid flows. Extending to 3D, viscous flow is an exciting direction, which can be enabled by allowing vortex strengths and sizes to evolve in time (Mimeau & Mortazavi, 2021). Secondly, our vortex evolution did not take into account the boundary conditions in a physically-based manner, hence it cannot accurately predict flow details around a solid boundary. Incorporating learning-based boundary modeling may be an interesting exploration. Thirdly, scaling our method to handle turbulence with multi-scale vortices remains to be explored. We consider two additional directions for future work. First, we plan to explore the numerical accuracy of our neural vortex representation to improve the current vortex particle methods for scientific computing. Secondly, we plan to combine our differentiable simulator with neural rendering methods to synthesize visually appealing simulations from 3D videos.”
>
> **Q4**: Would be great if the authors provide code.
>
> **A4**: Yes, we will release all code and data upon publication.
>
> **References:**
>
> [Hald, 1979] Hald, Ole H. “Convergence of Vortex Methods for Euler’s Equations. II.” SIAM Journal on Numerical Analysis 16, no. 5 (1979): 726–55.
>
> [Mimeau and Mortazavi, 2021] Chloé Mimeau and Iraj Mortazavi. "A review of vortex methods and their applications: From creation to recent advances." Fluids 6.2 (2021): 68.
>
> [Cortez, 1995] Ricardo Cortez. Impulse-based methods for fluid flow. No. LBL-37206. Lawrence Berkeley National Lab.(LBNL), Berkeley, CA (United States), 1995.

---

### Official Review · Reviewer_W2do · 2022-10-27

**Confidence:** 3
**Correctness:** 3
**Technical Novelty And Significance:** 3
**Empirical Novelty And Significance:** 4
**Recommendation:** 6

**Clarity, Quality, Novelty And Reproducibility:**

This paper is well written and easy to follow.

In terms of quality, this paper presents some novel empirical results, but the underlying theoretical model is standard to the literature.

**Strength And Weaknesses:**

Strength:
This paper proposes a fluid dynamics representation with differentiable vortex particles so that the observed particle densities and be used jointly with the underlying vortex model (the vorticity formulation of the 2D Navier-Stokes equation with the Biot-Savart kernel) to learn the particle trajectory and the vorticity-to-velocity mapping, i.e. an approximated Biot-Savart kernel. The empirical results are impressive compared with previous SOTA of the domain.

Weakness:
Since we exactly know the dynamics of the vortices, why do we need data to help to learn their trajectories? Given the initial distribution of the vortices, the following dynamics can be derived by solving the 2D Navier Stokes equation, right?

**Summary Of The Paper:**

This paper considers the modeling of fluid dynamics from limited observations of particle densities, using the vorticity formulation of the 2D Navier-Stokes equation. Specifically, the velocity field that drives the movement of the particles are inferred from the observable particle density over a time interval. The calculated velocity field is then modeled by the vortex model of the 2D Navier-Stokes equation with a learned approximation of the kernel of the Biot-Savart Law. The trajectory module of the vortices is trained so that the predicted velocity field matches the ones derived from the training data (the density image). Further, this paper considers the prediction of the evolution of the particle density based on the learned vortex dynamics.

**Summary Of The Review:**

The empirical results are impressive. However, it is not clear why the data are needed given that the underlying dynamics is already modeled by the 2D Navier Stokes equation. Is it for the purpose of estimating the viscosity parameter and the initial vortex distribution?

---

> ### Author Response · Authors · 2022-11-19
> **Response to Reviewer W2do**
>
> Thank you very much for your constructive and insightful review. Please find our response below:
>
> **Q1**: Since we exactly know the dynamics of the vortices, why do we need data to help to learn their trajectories? Given the initial distribution of the vortices, the following dynamics can be derived by solving the 2D Navier Stokes equation.
>
> **A1**: You are right in that, if the initial distribution of the vortices is given, then their trajectories are uniquely determined. **However, please note that our model takes as input a single RGB video, and the initial condition is unknown.** Therefore, we need to infer the initial condition from the video frames; and we achieve that through learning the full trajectory.
>
> An alternative to “learning initial condition through trajectory” is to learn the initial condition directly. However, we find that the former option is more computationally tractable and effective, if we want to fully exploit the input video. To see this, suppose we have 100 training frames in the video, and the goal is to infer the initial condition at frame #1. If we wish to directly optimize the initial condition using the last frame, we need to simulate from frame #1 all the way to #100, compute the loss and backpropagate. Unrolling such a long sequence for each training iteration (1) takes a long time, (2) leads to noisy gradients, and (3) is practically infeasible due to memory constraints. On the other hand, learning through the whole trajectory allows us to address these challenges by using smaller sliding windows in time (e.g., simulating only 3 frames at a time) and aggregating the dynamics information throughout the whole video. In Appendix F of our revised manuscript, we further elaborate on this argument, and show a quantitative comparison experiment of our technique vs. learning initial conditions only. We demonstrate that learning the initial condition only would fail to properly recreate the fluid motion, while learning through trajectory performs the task convincingly.
>
> Finally, we note that inferring the initial condition is non-trivial. We show this in the newly added Appendix B, where we compare our method to a standard grid-based differentiable fluid simulator (we implement a classic fluid simulator [Fedkiw, Stam and Jensen, 2001] in PyTorch, and a similar implementation and experimental setup has been adopted and validated by [Hu et.al., 2020]). We use the grid-based differentiable fluid simulator to optimize the initial condition to fit the input video. We perform the experiment on both real and synthetic videos. For both cases, we observe that the grid-based differentiable simulator can only faintly capture the large-scale movements, and yet the structures of the eddies are notably inaccurate, due to the lack of physical structures in its representation. This suggests that our successful learning of the initial condition is enabled by our unique Lagrangian-based formulation. Please refer to the figures and detailed numerical analysis in Appendix B for more.
>
> **Q2**: It is not clear why the data are needed given that the underlying dynamics is already modeled by the 2D Navier Stokes equation.
>
> **A2**: The answer to this question is twofold. First, as discussed above, we infer the initial condition of the fluid system from observed video frames. Secondly, data helps to solve the 2D Navier-Stokes equations better. Although the equations are analytically given, solving them by discretization involves various approximations and inaccuracies (e.g., the mollified velocity kernel). In traditional fluid simulation, these velocity kernels are hand-tuned Gaussian kernels with fixed size, which works fine in generating nice-looking flow patterns, but are not necessarily true to the specific flow in the input video. To address this, we propose to learn the velocity kernel from the input video frames to more accurately capture the dynamics traits particular to the video. We illustrate its advantage with an added experiment in Appendix E, where we perform ablation studies and show how data-driven learning can visibly improve the accuracy of reconstructing and extrapolating the input videos.
>
> **References:**
>
> [Fedkiw, Stam and Jensen, 2001] Ronald Fedkiw, Jos Stam, and Henrik Wann Jensen. "Visual simulation of smoke." SIGGRAPH, 2001.
>
> [Hu et.al., 2020] Yuanming Hu, Luke Anderson, Tzu-Mao Li, Qi Sun, Nathan Carr, Jonathan Ragan-Kelley, and Frédo Durand. "Difftaichi: Differentiable programming for physical simulation." ICLR, 2020

---

### Author Response · Authors · 2022-11-19
**General Response to All Reviewers**

We thank our reviewers sincerely for their insightful and inspiring comments. Following their suggestions, we have revised our manuscript in the following ways (changes in the manuscript are highlighted in blue):
1. We’ve included more details of our method, including:
    - The hyperparameter $n$ for the number of vortex particles.
    - The initialization of vortex particle positions.
    - The initialization of vortex particle sizes $\Delta$ and strengths $\Omega$.
    - The computational cost of our method.
2. We’ve added discussions and clarifications on several important concerns.
    - The validity and qualifications of assuming time-invariant vortex strengths and sizes.
    - The influence of hyperparameter $n$ on learning.
    - Comparison with existing differentiable fluid simulation.
    - The importance of data-driven velocity kernels.
    - The importance of trajectory learning.
3. To support these discussions, we’ve added the following five experiments.
    - Comparison with a standard differentiable, grid-based simulator on both synthetic and real videos (Appendix B).
    - Additional benchmark testing against 4 baselines with detailed numerical analysis (Appendix C).
    - Learning the same underlying fluid motion using different choices of hyperparameter $n$ (Appendix D).
    - Additional ablation study: learning with or without data-driven velocity kernels (Appendix E).
    - Additional ablation study: learning trajectory versus learning initial configuration only (Appendix F).

We have also replied to each of our reviewers individually under the comment sections. Please refer to these comments for further discussions.

---

### Decision · Program_Chairs · 2023-01-20

**Decision:**

Accept: poster

**Justification For Why Not Higher Score:**

The method is tested on very small datasets, at least for a machine learning conference. I highly recommend authors to include more comparisons with at least an order of magnitude more data.

**Justification For Why Not Lower Score:**

Both theory and experiments check out, as noted by the reviewers.

**Metareview: Summary, Strengths And Weaknesses:**

This paper proposes to learn to simulate fluid dynamics using differentiable vortex particles. This allows for using the vorticity formulation for the 2D Navier-Stokes, to learn particle trajectories, with strong results. All reviewers agree on the contributions, and the importance of the results. At the same time, there is also criticism that the current method works with 2D settings only, at least that is shown in the paper, and the size of the datasets used is quite small.

**Note From Pc:**

if the above contains the word "oral" or "spotlight" please see: "oral" presentation means -> notable-top-5% and "spotlight" means -> notable-top-25%. As stated in our emails, we are disassociating presentation type from AC recommendations